# A general model unifying the adaptive, transient and sustained properties of ON and OFF auditory neural responses

**Ulysse Rançon**[1]*, **Timothée Masquelier**[1], **Benoit R. Cottereau**[1,2]

**1** CerCo UMR 5549, CNRS – Université Toulouse III, Toulouse, France, **2** IPAL, CNRS IRL62955, Singapore, Singapore

☯ These authors contributed equally to this work.
* ulysse.rancon@cnrs.fr, ulysse.rancon@gmail.com

## Abstract

Sounds are temporal stimuli decomposed into numerous elementary components by the auditory nervous system. For instance, a temporal to spectro-temporal transformation modelling the frequency decomposition performed by the cochlea is a widely adopted first processing step in today's computational models of auditory neural responses. Similarly, increments and decrements in sound intensity (i.e., of the raw waveform itself or of its spectral bands) constitute critical features of the neural code, with high behavioural significance. However, despite the growing attention of the scientific community on auditory OFF responses, their relationship with transient ON, sustained responses and adaptation remains unclear. In this context, we propose a new general model, based on a pair of linear filters, named *AdapTrans*, that captures both sustained and transient ON and OFF responses into a unifying and easy to expand framework. We demonstrate that filtering audio cochleagrams with AdapTrans permits to accurately render known properties of neural responses measured in different mammal species such as the dependence of OFF responses on the stimulus fall time and on the preceding sound duration. Furthermore, by integrating our framework into gold standard and state-of-the-art machine learning models that predict neural responses from audio stimuli, following a supervised training on a large compilation of electrophysiology datasets (ready-to-deploy PyTorch models and pre-processed datasets shared publicly), we show that AdapTrans systematically improves the prediction accuracy of estimated responses within different cortical areas of the rat and ferret auditory brain. Together, these results motivate the use of our framework for computational and systems neuroscientists willing to increase the plausibility and performances of their models of audition.

## Author summary

Responses to stimulus onsets and offsets are ubiquitous along the auditory pathway and bear significant behavioral importance for the proper discrimination of natural sounds. We propose a general and unified descriptive model that links ON and OFF responses

**Data Availability Statement:** Datasets used in this study are publicly available online and were used with respect to their original license. They can

respectively be found on the CRCNS website (https://crcns.org/data-sets/ac/ac-1/about), Zenodo (https://zenodo.org/records/7796574), and the Open Science Framework (OSF) website (https://osf.io/ayw2p/). All Python codes necessary to build computational models and train them are made freely accessible at the following Github repository, alongside instructions about how to pre-process the datasets: https://github.com/urancon/deepSTRF.

**Funding:** This study was supported by a grant from the Agence Nationale de la Recherche (ANR-21-CE28-0021, ANR PRC ReViS-MD, https://anr.fr/) and by a FLAG-ERA funding (Joint Transnational Call 2019, project DOMINO, https://www.flagera.eu/), both awarded to BRC. The funders had no role in study design, data collection and analysis, decision to publish, or preparation of the manuscript.

**Competing interests:** The authors declare that no competing interests exist.

to sensory adaptation, encompassing previous computational approaches. The model consists of a simple pair of parameterized temporal filters that produce a bipolar spectrogram. We use the framework of digital signal processing to derive its mathematical properties, and demonstrate that they accurately reproduce known features of offset responses. We further validate our model by integrating it into larger pipelines, and show a systematic improvement of the latter at fitting neural responses. Our approach emphasizes the need to benchmark a wide range of models on several datasets, in a field where harmonization is lacking. In addition to better explaining the brain, our model could also serve in deep learning models for common engineering tasks such as audio recognition.

## Introduction

In signal processing, increments and decrements in the intensity of a stimulus constitute, as much as the stimulus intensity itself, valuable features to encode for further analyses. The sensory systems of numerous animal and notably mammal species exploit these intensity changes to encode sparse representations of their inputs and thereby improve their efficiency [1]. While the parallel processing of light intensity increments ('ON' signals) and decrements ('OFF' signals) along the visual pathway is now well documented [2, 3]) and actually led to the development of a new type of bio-inspired sensors (event-based cameras, see [4]), less is known about how sound intensity increments and decrements are processed by the mammal brain. Originally observed in the brainstem of bats [5], auditory OFF responses have since been measured along the auditory pathway in a wide variety of animal species: in the cochlea and auditory nerve [6], brainstem [7, 8], midbrain [9, 10], thalamus [11], and cortex [12–14]. Although less prevalent than ON responses [15–17], their ubiquitous occurrence is considered to be due to both bottom-up inheritance [9, 18, 19] as well as *de novo* generation [10, 12], thus leading to a dual pathway [20, 21]. If only little is known about their origins, the most accepted hypothesis relies on a post-inhibitory rebound phenomenon due to ionic mechanisms ([22, 23], see [19] for a model at the network level). It is now clear that these responses have important consequences at the behavioral level, notably for sound duration perception [24], gap detection [25–27], and at a higher level for communication [18]. Recent optogenetic studies in mice notably established that suppressing offset responses resulted in a performance drop on sound duration discrimination [15] and sound termination detection [12] tasks.

Despite this now established importance of auditory offset responses, only a few computational models took them into account. While some of them suffered from a high level of complexity [19, 28], others were constrained to low-level processes [23], or were difficult to interpret in terms of biological mechanisms because they were based on deep-learning black boxes [29, 30]. Besides, even fewer studies have made the connection between OFF responses and adaptation, sustained responses, and ON responses, although they might not be independent from each other [31] and some auditory neurons display all these types of behaviour [9, 17, 27, 32]. For instance, [33] proposed a frequency-wise model of adaptation inspired from electrophysiological measurements in the inferior colliculus (IC) of anaesthetized ferrets. If their approach increased the response-fitting ability of a linear-nonlinear (LN) model, they did not test whether it generalized to other types of models, nor if it also provided better fits on data collected in other species. Also, this work focused only on the phenomenon of adaptation, even if their model is capable of extracting –without segregation– onsets and offsets. In this

vein, [34] showed that incorporating a spectrally-tuned short-term plasticity (STP) in a variety of LN-based models improves their neural fitting performances, but this work focused on ferret data and only used a single model family, without expliciting the role of onsets and offsets. In the study performed by [27], authors proposed a simple computational model of auditory ON and OFF responses with split pathways to explain pathological deficits of gap detection in ectopic mice. Despite their convincing results, their model was applied to raw sound level, and therefore cannot generalize to responses to separated frequency bands, nor to interactions between them.

To address these shortcomings, we propose here a new general model of ON-OFF neural responses and adaptation in the mammal auditory pathway. This model encompasses previous approaches and is implemented on a widely used tool for deep learning (PyTorch). Its properties are presented from a signal processing perspective. It is composed of two linear filters (one for ON and the other for OFF responses) that capture the sustained and transient properties of auditory inputs within each frequency band (see the "AdapTrans" model of auditory ON-OFF responses and adaptation section). We demonstrate that this model accurately reproduces previous biological findings such as the dependence of OFF responses on the stimulus fall time and on the preceding sound duration (see the 'Results' section). We also demonstrate that our filtering approach greatly improves neural response fitting performances of a large variety of models of the auditory pathway, going from simple linear models to state-of-the-art multi-layer convolutional neural networks. This is done across three datasets (collected in different mammal species and under different experimental conditions) to permit robust estimations of the performances and hence more reliable conclusions. All the data, models, processing and pre-processing codes are publicly available on our github repository (https://github.com/urancon/deepSTRF).

## Materials and methods

Because our modelling framework is an essential part of the present study, it is introduced here, before the 'Results' section. Therefore, in this section, we first describe our model of auditory ON-OFF responses and adaptation, and formally analyze its properties. Then, we present larger computational models of the auditory pathway that were combined to our pair of filters to predict neural responses datasets (audio stimuli and neural recordings) that were used in our study. Next, we provide details about the datasets (audio stimuli and neural recordings) that were used for this purpose. Finally, we described our methodology to characterize the performances on the neural response fitting task.

### "AdapTrans" model of auditory ON-OFF responses and adaptation

**Two filters to capture the onsets and offsets of auditory inputs.**   Neural responses in the mammal auditory cortex depend on both the sustained and transient properties of input sounds and rapidly adapt their firing rates to intensity modulations [35, 36]. In order to take into account these properties, our model is based on a pair of filters that we call *AdapTrans* (for "*Adaptation and Transients*") which efficiently computes ON and OFF responses to sound onsets and offsets. These filters also maintain a sensitivity to the raw amplitude of sounds in different frequency bands, as observed in biology [36, 37]. Inspired by a previous study which modelled visual processes in the retina [38] and similar to the model of auditory processing proposed by [33], our approach consists in partially high-pass temporal filtering operations on the cochleagram with frequency-dependent exponential kernels. However, instead of using only one set of filters, we use two in order to separately compute the

responses to the sound onsets and offsets, as it is done in the auditory cortices of rats [20] and mice [21]. In signal processing theory, our filters can be categorized as causal, first order, biphasic, and with infinite impulse responses (IIR). These IIRs (or "*kernels*") are shown Fig 1A and can be formulated as:

$$
\begin{cases}
h_{ON}[n] & = \delta[n] - C_{ON}w\sum_{d=1}^{+\infty}a_{ON}^{d-1}\delta[n-d] \\
\\
h_{OFF}[n] & = -w*\delta[n] + C_{OFF}\sum_{d=1}^{+\infty}a_{OFF}^{d-1}\delta[n-d]
\end{cases}
\tag{1}
$$

with $n$ the discrete time variable, $\delta$ the Kronecker delta function

$$
\delta[n] = \begin{cases} 1 & \text{if } n = 0 \\ 0 & \text{otherwise} \end{cases}
\tag{2}
$$

and $a \in [0, 1]$ a real-valued parameter relative to the time constant $\tau$ (in timesteps) of the exponential part of the kernel ($a = exp(-1/\tau)$), $w \in [0, 1]$ a real-value parameter, and $C$ a normalization factor such that elements of the exponential sum to 1. Specifically, for an infinite exponential part (i.e., IIR filter):

$$
\begin{aligned}
C \quad & = \frac{1}{\sum_{k=0}^{+\infty} a^k} \\
& = \frac{1}{\frac{1}{1-a}} \qquad (sum\ of\ an\ infinite\ geometric\ series) \\
& = 1 - a
\end{aligned}
\tag{3}
$$

Essentially, these kernels compute a weighted difference between the current value of the signal and its exponential moving average (EMA) in a recent past. With this formulation, the $a$ parameter is related to the time constant of the exponential: the closer to 1, the higher (i.e., slower) the time constant. Meanwhile, the $w$ parameter allows to control the ratio between current (stem) and past (EMA) signal values. Equivalently, it can be interpreted as the ratio between permanent and transient features that are computed by the filter: with a value of 0, the ON-response only accounts for the raw signal; with a value of 1 it only accounts for its derivative.

Given the equations above, our filters have the following properties (see Fig 1B): 1) sound onsets on the ON channel lead to the same amplitude as sound offsets on the OFF channel, 2) sound offsets on the ON channel lead to the same amplitude as sound onsets on the OFF channel, 3) sustained sounds lead to the same amplitude on the ON and OFF channels in permanent regime and 4) the output of the filters is null when there is no auditory input. Importantly, although our two filters do not introduce a bias towards either of the two polarities, they are asymmetrical and linearly independent from each other (see Fig 1A), as documented in several studies on the auditory pathway ([13, 29, 32]). In the next section, we further describe the influence of the filter parameters on their responses.

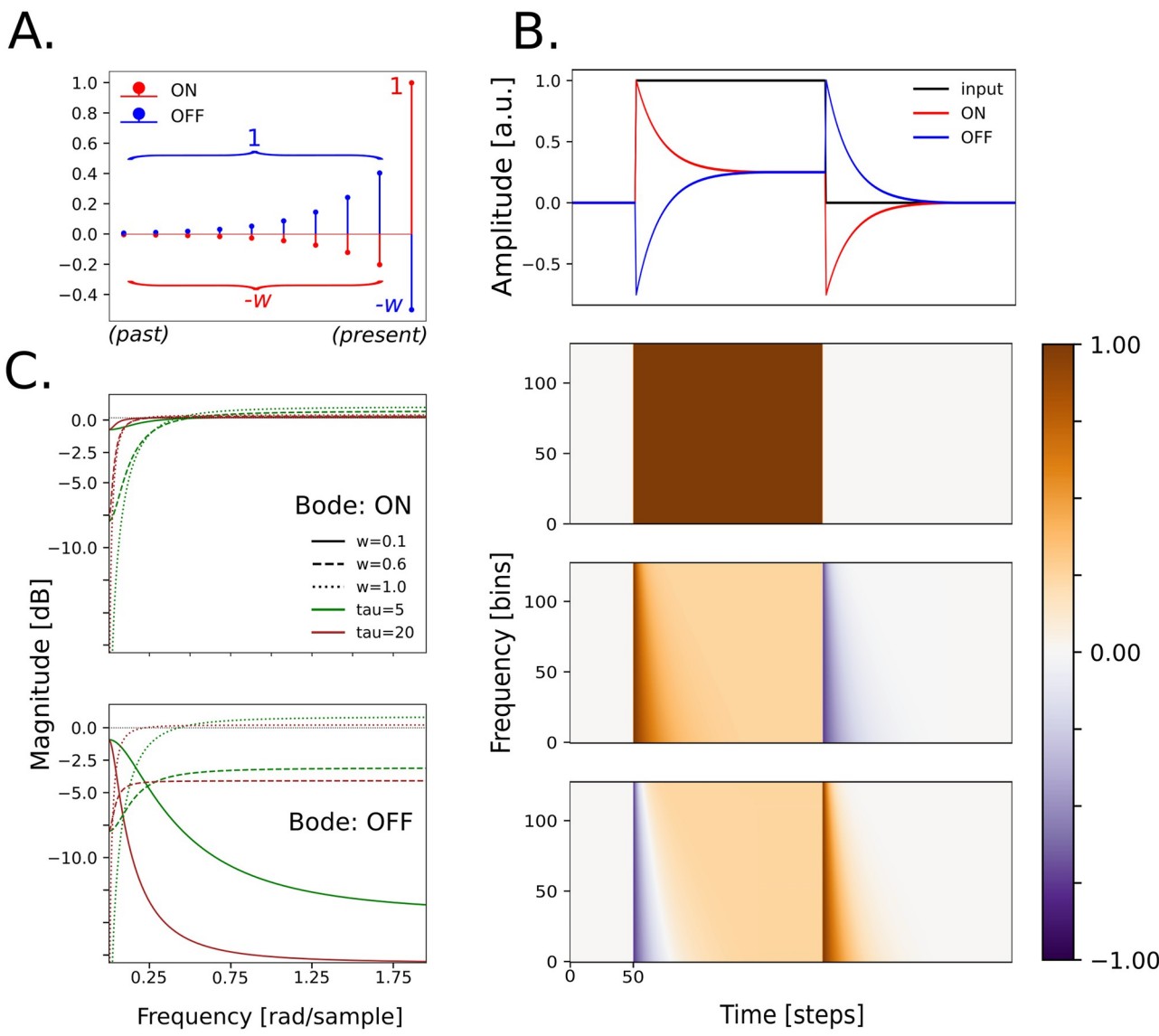

**Fig 1. Presentation of AdapTrans filters.** (**A**.) ON and OFF kernels of an AdapTrans filter of size 10, with $w = 0.5$ and $a = 0.6$. Most recent time-steps are represented on the right. Each kernel is temporally convolved with a cochleagram spectral band from left to right, thereby computing a weighted difference between the current value of the signal and an exponential average of its recent past. Importantly, the output of the OFF channel is not the opposite of the output of the ON channel, since both kernels are linearly independent. (**B**.) Example ON and OFF outputs for a dummy input spectrogram composed of 128 frequency bands. All frequency bands of the latter bear the same signal: a rectangle function with one onset, one permanent regime, and one offset *(top)*. The ON channel responds positively to the onset and negatively to the offset, while the OFF channel does the opposite. The ON channel has an initial onset response of 1 –the value of the step in the input signal– and a negative offset response of $-w$; similarly the OFF channel has a negative onset response of $-w$ and an offset response of 1. Importantly, both polarities share the same sustained activity. For this particular input, the permanent response in both channels equals $1 - w$. Time constants are logarithmically distributed along the frequency axis, therefore producing slower responses for lower cochlear bands *(bottom 2 sub-panels)*. (**C**.) Bode plots of the ON and OFF filters, for different sets of parameters $\tau$ and $w$.

**Parametric frequency analysis.** The transfer function of both filters can be obtained as the *Z*-Transform of their impulse responses (derivation in S1 Note):

$$\begin{cases} H_{ON}(z) & = \dfrac{1 - (a + w - aw)z^{-1}}{1 - az^{-1}} \\[3mm] H_{OFF}(z) & = \dfrac{-w + (1 - a + aw)z^{-1}}{1 - az^{-1}} \end{cases} \tag{4}$$

The frequency responses of our filters can be better characterized with their Bode diagrams which are shown on Fig 1C (see the S1 Note for the associated analytic formula). Both filters are generally high-pass. The cutoff frequency of the ON filter depends on the time constant $a$, while $w$ acts as gain tuner: the closer to 1 the higher the high-pass effect. With $w = 0$, the exponential part of the ON kernel vanishes, thereby reducing it to a single Kronecker delta function and leaving any auditory input unchanged. This is reflected in the Bode diagram with a flat magnitude-frequency curve. With $w = 1$, ON and OFF kernels are opposite and thus have the same frequency response, which is not efficient from a computational point of view. Interestingly, the OFF kernel can also turn into a low pass-filter for low $w$. This analysis confirms that the proposed family of paired filters actually highlights the transient properties of input acoustic stimuli. As hinted above, it provides a simple interpretation to both parameters, $a$ regulating the "adaptation" part (i.e., the time constant with which to compute the exponential average of past inputs) and $w$ the "transient" part (i.e., the relative importance of current inputs with respect to previous ones).

**A frequency-wise application.** In our framework, we apply the pair of filters independently on each frequency band of the cochleagram. This choice is mainly motivated by electrophysiological observations of ON and OFF responses in animal models ([12], see also our results in the next section). Another argument for this multi-frequency filtering is that auditory neurons can display both different frequency and polarity tuning [39], and that spectral tuning has been shown to better accounts for their responses [34]. Therefore, we used different sets of AdapTrans parameters ($w$, $a_{ON}$, $a_{OFF}$) for each cochlear frequency, and treat them as learnable optimization parameters in our experiments, fitted jointly with the others parameters of the neural response model backbone (see Integration within larger models of audition). Given that auditory neurons tend to have larger time constants for lower frequencies [36], we set initial values (i.e., pre-optimization) of AdapTrans time constants to follow a biologially-plausible logarithmic function derived from experimental data [33]:

$$\tau(f) = 500 - 105 \log(f) \tag{5}$$

with $\tau$ in milliseconds and $f$ in Hertz. We initialized $w$ to 0.75 for all frequencies. This value close to 1 is justified by the fact that onset-sustained-offset neurons in the auditory cortex display small, yet nonzero, sustained activity upon presentation of a sound.

**Implementation.** In the present study, for computational efficiency and parallelization on modern GPU hardware, we truncated the IIR, such that our implementation of the AdapTrans filters was based on a kernel with a finite number of elements. Its length was equal to $3 \times \tau_{max}$ + 1, $\tau_{max}$ being the time constant of the lowest cochleagram frequency band (see Fig 1A). In this case, approximately 95% of the exponential part of AdapTrans is properly represented. As indicated above, we normalized the exponential part by computing $C$ such that the terms respectively sum to 1 and $-w$ for ON and OFF polarities, respectively. We padded the input spectrogram to its left (past) in 'replicate' mode (i.e., using the left-most value, see PyTorch documentation) before the convolution operation in order to avoid any downsampling along

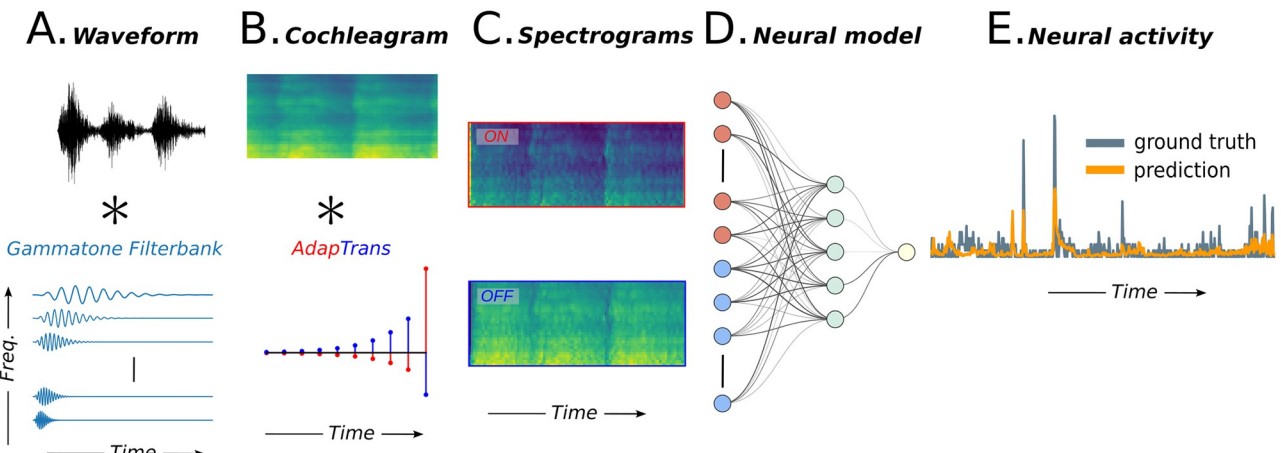

**Fig 2. Overview of the proposed processing pipeline.** (**A**.) Its inputs are given by a spectrogram representation of the sound stimulus (e.g. obtained from a Gammatone filterbank). (**B**.) Two AdapTrans filters are convolved to each frequency band along the temporal dimension, effectively accounting for transient and permanent features of the stimulus. (**C**.) The filter outputs consist of a 2-channel, ON-OFF spectrogram that is further processed by conventional models as found in the litterature. (**D**.) Such models can be separated into several classes, depending on whether they are based on a single and large spectro-temporal receptive field (STRF), a cascade of convolutions with small-sized kernels (CNNs), or recurrent neural networks (RNNs)(the architectures of the models used in this study are shown in Fig 3). (**E**.) All models output the predicted neural activity as a time series that can be compared to a groundtruth recording.

the time dimension. Our repository (available at https://github.com/urancon/deepSTRF/) contains an easy-to-use dedicated PyTorch class for AdapTrans.

**Integration within larger models of audition.** We explain here how our framework can be easily integrated into larger models of auditory processing, going from simple and gold-standard linear models to state-of-the-art convolutional neural networks. This integration is illustrated in Fig 2. Assuming an input sound waveform converted into its spectro-temporal representation in an initial processing step (e.g., using a gammatone filter bank), our Adap-Trans filters are applied on each frequency band (see above). The output of this filtering process is a 2-channel, ON-OFF spectrogram with transient and adaptive sustained activities. After being passed through a rectified linear unit (ReLU, or "*half-wave rectification*"), this tensor can then be fed to standard auditory neural response models, such as Linear (L) or Linear-Nonlinear (LN) by doubling the number of input channels, in order to be compatible with the ON and OFF channels of AdapTrans. As a result, the number of learnable parameters in the input layer of the response model is doubled, which is not an issue because in multi-layer neural network algorithms like deep Convolutional Neural Networks (CNN), the input layer only constitutes a negligible fraction of the total number of learnable parameters in the model. Most importantly, in our framework, these added parameters remain fully interpretable (weights for each frequency bin become weights for each time-frequency-polarity bin). Nevertheless, to evaluate as precisely as possible the effect of AdapTrans on model performances, and disentangle it from the effect of supplementary parameters, we divided by a factor of two the number of hidden units in NRF and DNet backbones, thereby bringing back their total parameter counts to the same level as their control counterpart (i.e. without AdapTrans, raw spectrogram as single channel). The corresponding numbers of parameters are provided in Table 1.

Additionally, if the initial distribution of the ($a_{ON}$, $a_{OFF}$, $w$) parameters of AdapTrans (see Eq 5) is drawn from experimental data, it might nonetheless not be optimal for the specific neural unit under study. We thus decided to jointly optimize these parameters alongside the

**Table 1. Technical details for each model investigated in this study.** The hyperparameters *F*, *T* and *H* respectively correspond to the number of frequency bins spanned by each convolutional layers, to the number of time bins and to the number of hidden units before readout. For a given model, these hyperparameters could vary between datasets because time-steps differed (e.g., 5 ms for NS1 and 10 ms for NAT4). As a result, the same model could have a variable number of learnable parameters, depending on the dataset it was trained on. To permit a fair comparison, all models had access to the same temporal span for a given dataset.

| Model \ Dataset | | NS1 (dt = 5ms) | | | | Wehr (dt = 5ms) | | | | NAT4 (dt = 10ms) | | | |
|---|---|---|---|---|---|---|---|---|---|---|---|---|---|
| Backbone | Prefiltering | F | T | H | #params | F | T | H | #params | F | T | H | #params |
| L / LN | None / IC adaptation | 34 | 41 | 1 | 1,395 | 49 | 21 | 1 | 1,030 | 18 | 21 | 1 | 379 |
| | AdapTrans | | | | 2,891 | | | | 2,206 | | | | 811 |
| NRF | None / IC adaptation | 34 | 41 | 20 | 27,961 | 49 | 21 | 20 | 20,661 | 18 | 21 | 20 | 7,641 |
| | AdapTrans | | | 10 | 28,023 | | | 10 | 20,768 | | | 10 | 7,655 |
| DNet | None / IC adaptation | 34 | 5 | 20 | 3,502 | 49 | 5 | 20 | 5,002 | 18 | 5 | 20 | 1,902 |
| | AdapTrans | | | 10 | 3,552 | | | 10 | 5,099 | | | 10 | 1,906 |
| 2D-CNN | None / IC adaptation | 6 | 15 | 90 | 36,274 | 7 | 6 | 90 | 39,694 | 3 | 7 | 90 | 15,484 |
| | AdapTrans | | | | 37,276 | | | | 40,261 | | | | 15,748 |

parameters of the downstream models, through gradient descent, and for each neuron. In practice, this general approach permits the optimizer to find the best set of AdapTrans parameters in order to explain neural activity, encompassing a large variety of cases (including the identity transform, which would let the raw spectrogram unchanged for $w = 0$, or the derivative for $w = 1$).

## Computational neuron models

We review here the computational response models used in this study. Because the most common models (L and LN) are well established and were extensively described in previous modeling works, we only focus here on their main properties. Unless stated otherwise, we do not parameterize the spectro-temporal kernels of these network, nor regularize them using weight decay. Furthermore, for a smoother learning process and better convergence, we also introduced Batch Normalization (BN) [40] in our models. BN is a simple and widely used form of normalization in deep learning that stabilizes gradient descent and increases model performances, while respecting linearity (thus after training it can be absorbed in the preceding convolutional or fully connected layer). All model architectures are shown in Fig 3, and Table 1 compiles the hyperparameters used on each dataset.

**Auditory periphery.** To facilitate present and future comparisons with previous methods, and to limit as much as possible the introduction of biases due to different stimulus pre-processings, we directly used the cochleagrams provided in each of the associated datasets. They were all obtained following similar principles, that is, a short-term spectro-temporal decomposition from windowing functions, followed by a compressive nonlinearity. However, despite their overall resemblance and for the sake of completeness, we briefly review here the main computations which were performed (more details can be found in the associated papers). Table 2 partly summarizes the waveform-to-spectrogram encoding for each dataset.

In the NS1 dataset [41], 10 ms Hanning windows (overlap: 5 ms) were used to compute the short-term amplitude spectrum of auditory stimuli which was subsequently transformed into a spectrogram using a set of 34 mel filters logarithmically spaced along the frequency axis (500–22,627 Hz). Finally, a logarithmic function was applied to each time-frequency bin, any value below a manually-defined threshold was set to that threshold, and the cochleagrams were normalized to zero mean and unit variance across the training set.

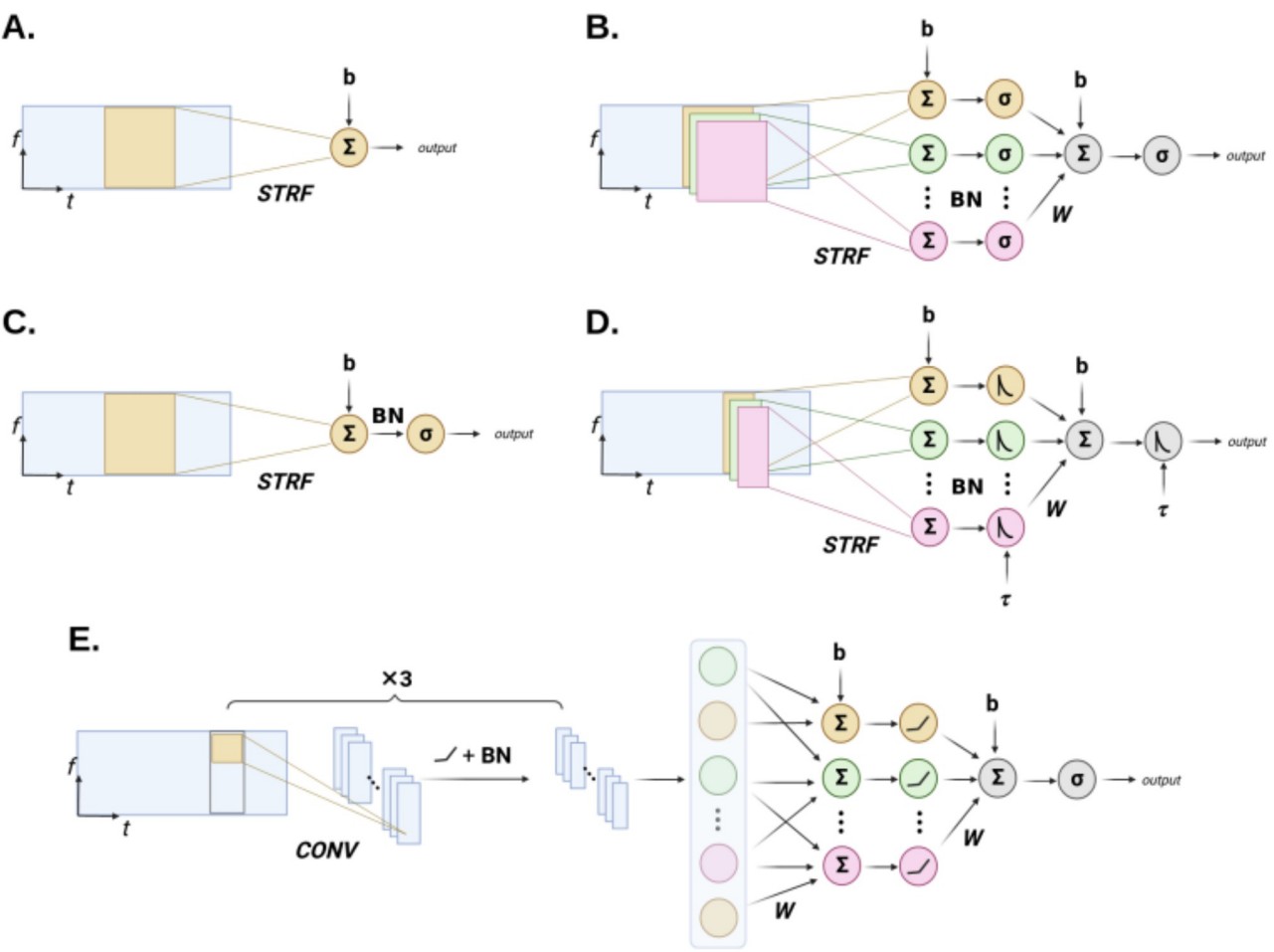

**Fig 3. Schematic expliciting the architecture of the different auditory response models used in combination to AdapTrans.** *STRF*: Spectro-Temporal Receptive Field(s). *CONV*: Convolution. *BN*: Batch Normalization. See main text and associated papers for a detailed description of each model. (**A**.) L model. (**B**.) NRF model. (**C**.) LN model. (**D**.) DNet model. (**E**.) 2D-CNN model.

In the NAT4 dataset [42], sound waveforms were converted into cochleagrams with a gammatone filterbank of 18 filters whose central frequencies were logarithmically distributed from 200 Hz to 20 kHz. After a log-compression stage, the temporal resolution was downsampled to 10 ms.

Finally, stimulus spectrograms from the Wehr dataset ([43, 44]) were obtained using a short-term Fourier transform with a Hamming window, followed by a log function. The resulting cochleagrams have a temporal resolution of 5 ms, while the frequency axis is discretized into 49 logarithmically-scaled spectral bins.

**Linear model (L).** The canonical Linear (L) model consists of a spectro-temporal window spanning all frequency bands and a large set of temporal delays that is convoluted over the temporal dimension of the stimulus spectrogram. The output of this model at each time step (i.e., the time-step of the most recent delay) is simply a linear projection of past spectro-temporal bins into a scalar value, to which a bias term is added to account for spontaneous activity. The set of weights associated with each input coefficients is also known as the *Spectro-Temporal Receptive Field* or STRF. Therefore, for a spectro-temporal window spanning F frequency bins and T delays, the number of parameters is $F \times T + 1$. Different parametrizations exist to

**Table 2. Review of the datasets used in this study.** Mean $CC_{max}$ was computed accross neurons and stimuli for responses with more than 1 repeat. Please refer to main text or their original study for additional details.

| | | Dataset | | | |
| --- | --- | --- | --- | --- | --- |
| | | **NS1** | **Wehr** | **NAT4** | |
| | | | | **PEG** | **A1** |
| **Literature** | original paper | [41] | [44] | [42] | |
| | others | [50, 52] | [53] | [30] | |
| **Recordings** | animal model | ferret | rat | ferret | |
| | behavioural state | anesthetized | anesthetized | awake | |
| | brain areas | A1, AAF | A1 | PEG | A1 |
| | # valid neurons | 73 | 21 | 339 | 777 |
| | mean $CC_{max}$ | 89% | 96% | 59% | 61% |
| **Stimuli** | sound type | natural | | | |
| | duration | 5 s | 7.5–15 s | 1 s | |
| | # sounds | 20 | from 3 up to 63 | 595 | |
| | # repeats | 20 | from 1 up to 25 | 1 (577 sounds), 18 (20) | |
| | time bins | 5 ms | 5 ms | 10 ms | |
| | # frequency bins | 34 | 49 | 18 | |

reduce this number of free parameters [45–47], which we did not adopt to illustrate the performances of the simplest implementation of this model. For the same reasons, we did not apply any regularization technique like ridge regression [44, 48] or L1 penalty [49, 50] by setting weight decay to 0. These methods allow to mitigate the high-frequency patterns appearing during optimization on unregularized STRFs, but require computationally intensive and often time-consuming hyperparameter tuning. Furthermore, we noticed that they did not necessarily prove beneficial in terms of performance in our setup, in line with the results reported by [30]. This observation could be due to our usage of gradient descent (see [51]), whereas previous literature tended to use order-0 optimization algorithms like boosting [45, 46] to fit these simple models. For this model and each spectrogram prefiltering condition (i.e., none, IC Adaptation, AdapTrans), we only used batch normalization (BN) after STRF weights when it proved beneficial to performance.

**Linear-nonlinear model (LN).**   The LN model differs from the L model in that the output of the convolution is passed through a static nonlinearity in the form of a sigmoid:

$$y = \sigma(x) = \frac{1}{1 + exp(-x)} \tag{6}$$

where $x$ is the output of the Linear model and $y$ is the output of the nonlinearity. Other forms of activation functions are commonly used, such as 4-parameter parameterized sigmoids [33, 50], or double exponentials [30, 45], but our preliminary results with the latter did not necessarily yield better results than with the standard sigmoid function. Early experiments showed the importance of using BN in conjunction with nonlinearities. We thus systematically incorporated it between STRF weights and output activation function, for all conditions of this model backbone.

**Network receptive field (NRF).**   This model, proposed in [41], extends the LN model by replacing its unique spectro-temporal weighting windows by several. After a pass through a standard sigmoid activation, the features extracted by each of these channels are then combined into a single output scalar forming the final prediction at the current timestep. With its multi-filter paradigm, authors argued that this model fitted much better actual

electrophysiological recordings, due to the fact that auditory neurons react to several spectro-temporal patterns, and not just one. To follow the LN model (see above) and make its strict multi-filter extension, we also introduce BN between input weights and the hidden activation function.

**Dynamic network (DNet) model.**   [50] further extended the NRF model by constraining its hidden and output units to follow a recurrent, exponentially decaying relationship over time, similar to a non-spiking Leaky Integrate-and-Fire (LIF) neuron. Authors showed that this replacement of the sigmoid activation by a simple stateful dynamic observed in biology, allows to reduce the span of the STRF windows to a more biologically plausible range, without sacrificing performance. Implementation-wise, removing the spiking condition allows to emulate the leaky recurrence by simply convolving an exponential kernel along the time dimension of each layer's output. We parametrize the exponential kernel the same way as AdapTrans filters, and let automatic differentiation learn the time constant, which we express as follows for numerical stability:

$$\frac{1}{\tau} = \frac{1}{1 + d^2} \tag{7}$$

We mark a difference with these authors, in that we allow the network to learn a different time constant for each hidden unit, instead of a single one that is shared accross the layer. Similar to both LN and NRF models, we employed BN in the first layer, between input weights and hidden units.

**2D Convolutional neural network (CNN).**   Also based upon convolution operations on the stimulus spectrogram, this last model differs from the preceding in that it is a fully-fledged, deep neural network, with a larger number of stacked convolutional layers of small kernels (i.e. that do not span the entire range of frequencies). Introduced by [30] among other CNN-based models, it displayed superior performances on the task of elecrophysiological response fitting, despite its higher number of learnable parameters. Our PyTorch re-implementation aimed at being as close as possible from the original architecture: a feature extraction backbone constituted in a series of three 2D convolutions alternating with nonlinear activations, followed by a prediction head composed of two fully connected layers. Nevertheless, we added a minor update by also incorporating BN between each convolution and nonlinear activation (LeakyReLU with a negative slope of 0.1), as it constitutes a well appreciated solution to mitigate overfitting among the deep learning community [40]. Similar to the original paper, we chose to maintain 10 hidden channels within the convolutional backbone, and 90 hidden units in the last layer, the size of penultimate layer being determined by flattening the downsampled spectrogram out of the convolution backbone.

## Sound datasets

To demonstrate the generalization ability of our approach, we chose audio data collected in different cortical areas (ferret A1, ferret AAF, ferret PEG, rat MGB) and at different temporal resolutions (1, 5 and 10 ms). These data come from recent studies ([41–43]) and are freely available on the internet. We only report here the main steps for their acquisition and preprocessing (see also Table 2 for an overview of their main characteristics). More details can be found in the corresponding papers. The preprocessed data, ready for Pytorch development, are all freely available on our GitHub.

**NS1 dataset.**   This dataset comprises single-unit extracellular electrophysiological recordings performed in the primary auditory cortex (A1) and anterior auditory field (AAF) of 6 anesthetized pigmented ferrets exposed to natural stimuli. Natural sound clips (n = 20)

included either birdsong, ferret vocalizations, human speech and environmental noises (e.g. wind, water). Each clip last 5 s and was presented 20 times to each animal at a sample rate of 48,828.125 Hz. A Klustawik-based spike-sorting algorithm [54] isolated a total of 73 single units that matched a certain noise ratio threshold, allowing the construction of their peri-stimulus time histograms (PSTH) by first counting spikes in 5 ms temporal windows, then averaging over repeats, and smoothing by convolution with a Hanning window. This yielded suprathreshold (i.e. firing probability) response profiles with a temporal resolution of 5 ms for each unit and sound clip.

Stimuli were first processed into a 34 band mel spectrogram of 5 ms time bins with frequencies ranging from 500 to 22,627 Hz. The log of each time-frequency bin was then computed, and values below a fixed threshold were set to that threshold. The resulting cochleagrams were finally normalized to zero mean and unit variance.

This dataset is available online on the Open Science Framework (OSF) website (https://osf.io/), at the repository associated to its original article [41]. Please refer to the latter for more details on the acquisition and preprocessing of the data.

**NAT4 datasets.** The following dataset [55] was acquired from the primary auditory cortex (A1) and secondary auditory field (PEG) of 5 awake ferrets exposed to a wide range of natural sound samples. Spiking activity was collected extracellularly through micro-electrode arrays (MEA) and single- and multi-units were isolated from raw traces using the Kilosort 2 spike sorter ([56]). In total, 777 auditory-responsive units were identified in A1 and 339 in PEG.

All of the 595 stimuli were 1 s long, and were presented with a 0.5 sec interval of silence. 15% were congenital vocalizations and noises recorded inside the animal facility, while the remaining 85% were taken from a public library of human speech, music and environmental sounds [57]. 577 of these sounds were repeated only once, and 18 were repeated 20 times. For each neuron and stimulus clip, we removed stimulus-response pairs with null PSTHs (i.e., without any spike in all response trials). These data are available in online open-access on the Zenodo platform (https://zenodo.org/), on a dedicated repository associated to its original paper [42].

**Wehr dataset.** In this last dataset ([43, 44]), pure tones and natural stimuli were presented to anesthetized Sprague Dawley rats while membrane potentials of neurons in their primary auditory cortex (A1) were recorded using a standard blind whole cell patch-clamp technique, in current-clamp mode ($I = 0$, sampling frequency: 4,000 Hz). Action potentials were pharmacologically prevented by the administration of a sodium channel blocker, therefore allowing large PSPs at most. For each of the 25 cells recorded in this study, the frequency tuning curve was determined thanks to the presentation of short pure tone stimuli (20 ms duration with a 5 ms cosine-squared ramp or 75 ms a 20 ms ramp) which were sampled and delivered at 97.656 kHz in a pseudo-random sequence. Natural sounds with various durations (7.5–15 sec with 20 ms cosine-squared ramps at onset and offset) originally sampled at 44.1 kHz were upsampled and delivered at 97.656 kHz. These natural stimuli were a selection of 122 commercially available clips of environmental noises and animal vocalizations, and covered frequencies from 0 to 22 kHz. Depending on neurons and experiments, these natural stimuli were repeated up to 25 times. Recorded neural responses in this dataset are characterized by very low variability and are therefore very reliable. As a result, raw and normalized correlation coefficients reported on this dataset are very similar.

Because of their nature compared to the other datasets (i.e., membrane potentials vs spikes), response traces could be subject to drift. These recording artifacts are often meaningless and difficult for models to bypass, so we detrended responses linearly, which resulted in improved fitting performances, especially for simpler models.

Three neurons were reported to be unresponsive to sound stimuli in [44] and we did not include these neurons in our analyses. We also discarded another one that significantly lacked data, bringing the total number of units used in our study to 21 for this dataset.

Similarly to previously described data, stimulus spectrograms resulted from a simple short-term Fourier transform (STFT) in which the frequency axis was logarithmically discretized into $F$ = 49 spectral bands (12 / octave); temporal resolution was set to $dt$ = 5 ms for our analysis. The resulting energy density spectrum of the sound pressure wave was passed to a log-compression function and then further multiplied by a factor 20.

This dataset is freely available online on the Collaborative Research in Computational Neuroscience (CRCNS) website (https://crcns.org/), and constitutes the first half of the "CRCNS-AC1" dataset. More details are available in its original article [44].

## Task and evaluation of performance

**Optimization process.** All models, including AdapTrans and the backbone, were trained using gradient descent and backpropagation, with AdamW optimizer ([58]) and its default PyTorch hyperparameters ($\beta_1$ = 0.9, $\beta_2$ = 0.999). We used a batch size of 1 for NS1 and Wehr datasets, which have a limited number of training examples, and a batch size of 16 for both NAT4 datasets, which have consequently more. The learning rate was held constant during training and set to a value of $10^{-3}$, as we empirically found that these values led to better results. We explored different strategies to reduce overfitting in our modelling, e.g. by using weight decay (L2 regularization), Dropout or data augmentation (TimeMasking and FrequencyMasking). As none of these strategies significantly improved our results, we did not consider them further. At the completion of each training epoch, models were evaluated on the validation set and if the validation loss had decreased in comparison to the previous best model, the new model was saved. Models were trained until there were no improvement during 50 consecutive epochs on the validation set, at which point the training was stopped, the last best-performing model was saved and evaluated on the test set.

This unified approach for implementing and optimizing the parameters of each of the models (i.e., using the same regularization method, fitting approach, number of cochleogram channels, . . .) allows a fair comparison between them (and also between models equipped with AdapTrans or not). Indeed, as all the models (L, LN, NRF, DNet and 2D-CNN) were constructed using exactly the same pipeline, it implies that a model with higher neural fitting performances is genuinely better. Note that this homogenisation strategy necessarily introduced differences between our general pipeline and those of the studies that originally described these models.

**Correlation coefficients between recorded and predicted responses.** The neural response fitting ability of the different models has been reported using the raw correlation coefficient (Pearsons' $r$), noted $CC_{raw}$, between the model's predicted activity $\hat{r}$ and the ground-truth PSTH $r$, which is the response averaged over trials $r_n$:

$$r = \frac{1}{N}\sum_{n=1}^{N} r_n \tag{8}$$

$$CC_{raw} = \frac{Cov(r, \hat{r})}{\sqrt{Var(r)Var(\hat{r})}} \tag{9}$$

with $Cov$ and $Var$ operators performed on the temporal dimension. However, due to noisy signals and limited number of takes, perfect fits (i.e. $CC_{raw}$ = 1) are impossible to get in practice.

As a result, in order to give an estimation of the best reachable performance given neuronal and experimental trial-to-trial variability, several metrics have been proposed, such as the normalized explained signal power [44, 59] or the normalized correlation coefficient $CC_{norm}$, as defined in [60] and [61]; we report the latter in this paper. Namely, for a given optimization set (e.g., *train*, *validation* or *test*) composed of multiple clips of stimulus-response pairs, we first create a long sequence by temporally concatenating all clips together. Then, we evaluate the signal power $SP$ in the recorded responses as:

$$SP = \frac{Var(\sum_{n=1}^{N} r_n) - \sum_{n=1}^{N} Var(r_n)}{N(N-1)} \tag{10}$$

which finally allows us to compute the normalized correlation coefficient:

$$CC_{norm} = \frac{Cov(r, \hat{r})}{\sqrt{SP \times Var(\hat{r})}} \tag{11}$$

In the extreme case of only one trial being available, we set $CC_{raw} = CC_{norm}$, corresponding to the best-case scenario of a fully repeatable recording uncontaminated by noise, therefore preventing any overestimation of performances by giving a lower bound of the latter, in absence of data.

**Coherence function.**   To better quantify the contribution of our approach in the frequency domain, we computed the coherence values between the predicted and actual neural responses (see e.g., [44]). This metric is defined by:

$$C_{r\hat{r}}(f) = \frac{|G_{r\hat{r}}(f)|^2}{G_{rr}(f) G_{\hat{r}\hat{r}}(f)} \tag{12}$$

where $r$ and $\hat{r}$ are the neural responses and their predictions. $|G_{r\hat{r}}(f)|^2$ is the squared magnitude of their cross-spectral density, and $G_{rr}(f)$ and $G_{\hat{r}\hat{r}}(f)$ their respective auto spectral densities. Here, $f$ spans frequencies going from 0 to the Nyquist frequency associated with the sampling rate. For each frequency, the coherence takes values between zero (no correlation between measured and estimated response) and one (perfect correlation at this frequency). Coherence was computed using the Welch's method available in SciPy library [62], with segments of 500 ms duration in order to capture the long-range temporal dependencies and contextual effects observed in auditory neurons [53]. For coherence plots averaged across neurons for a given model and dataset, we also provided an upper bound value by computing the average coherence between the PSTH obtained from one half of the response trials and the one obtained from the other half, for up to 126 different splits [60]. Similar to our method of calculation of the normalized correlation coefficient above in the case of single trial data, we set the coherence upper bound to the worst-case scenario of 1 at all frequencies, in order to avoid any overestimation of model performances.

**Cross-validation methodology.**   For the NS1 and Wehr datasets, neural recordings were split into training (70%), validation (10%) and test (20%) sets. For all the measurements of the NAT4 dataset (i.e., for units in A1 and PEG), we followed the same sets as in [30]: a training and validation set of 577 stimulus-response pairs with only 1 repeat, and a test set of 18 stimulus-response pairs with 20 repeats. Here, the validation set was constituted of 20% of the total "trainval" set, and the training set of the remaining 80%. As indicated in the 'Optimization process', the model is fitted on the training set for a limited number of epochs. At the end of each epoch, the loss over the validation set is computed, and the model with the lowest validation loss is saved at the end of the fitting procedure. The number of training epochs was determined manually such that more epochs would not further decrease the validation loss. Finally,

the saved model was evaluated on the test set. This procedure was repeated 10 times for different train-valid-test data splits of NS1 and Wehr datasets and model parameters initializations, and the test metrics were averaged across splits. In the case of NAT4 datasets, because the number of stimulus-response pairs and neurons are considerably greater, risks of overfitting to a specific data split are much lower, and therefore we only report the performances for one random seed.

Note that this model validation method differs from the one employed in [50], which is not a "cross-validation" *per se*. In this study, authors kept a fixed, (i.e. always the same) subset of data for testing, and used the rest for training and validation. The test set was held-out during the process of model development, but because of the very small dataset size on NS1 (20 stimulus-response pairs), we found that this methodology was not robust to the selection of the test set, which could result in overestimated performances (see S1 Table).

### Reproducibility

Our simulations were done in Python using the popular automatic differentiation library PyTorch. Upon publication of this article, we will make our code freely available on Github at the following address: https://github.com/urancon/deepSTRF. A code example is provided in S1 Code, to showcase its simplicity and encourage other researchers to build upon it. Jobs required less than 2 GiB and were executed on Nvidia Titan V GPUs, taking 10 to 15 hours on NS1, depending on the complexity of the model.

Electrophysiology datasets openly available often come in a variety of formats that each necessitates a specific pre-processing. In addition, these pre-processing are performed using different tools and software. Because this lack of harmonization can induce errors and biases, there have been a few attempts to federate the scientific community around data hubs such as the Neural Prediction Challenge (https://neuralprediction.org/npc/index.php) or software toolkits like the NEMS library (https://github.com/LBHB/NEMS/tree/main). In line with these previous attempts, we provide all the models and the scripts to train them using a unified pipeline. We also provide more user-friendly PyTorch Dataset classes for each source of data used in this study. By building this repository, our constant goal has been to make an easy-to-use software material that is as plug-and-play as possible. We hope that the code architecture we have adopted will inspire other researchers by easing the development of future work, and ultimately contribute to make the bridge between experimental and theoretical research.

## Results

The aim of this work is to propose a model of auditory ON/OFF responses and adaptation, which can reproduce actual neural responses and properties measured from electrophysiological recordings in a wide range of animals, brain areas, and stimulus types.

### Dependence of OFF responses on stimulus fall time

Several studies have empirically demonstrated a dependence of offset responses on the stimulus fall time, in a variety of animal species and in different cortical areas: mouse MGB and AAF [12], rat AAF, A1 and VAF [22]. This property of offset responses is well captured by our model as abrupt down steps in the input stimuli elicit higher OFF responses than slow descending ramps of the same amplitude.

To prove it, we simulated the AdapTrans OFF response to a single-channel stimulus composed of a ramp of variable duration (see Fig 4A). We systematically computed the maximal offset response at sound termination as a function of fall time, and we searched for the $a_{OFF}$ and $w$ parameters that yielded the best fit with [12] data (see Fig 4A). Beyond the validation of

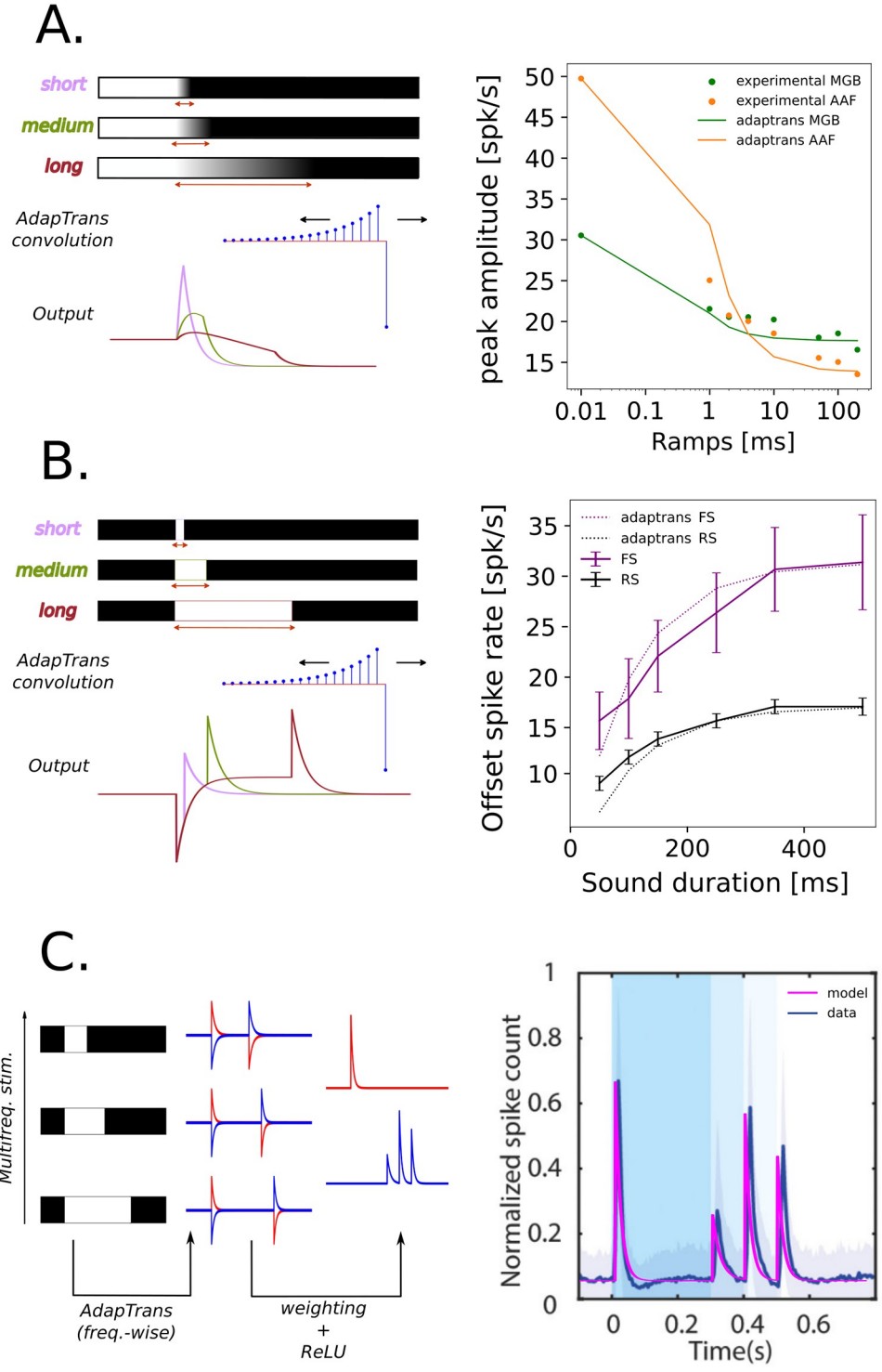

**Fig 4. Comparison between OFF responses predicted by our model and neurobiological data.** Experimental measurements were performed in the mouse medial geniculate body (MGB) and anterior auditory field (AAF)(see [12]). Despite a lesser effect of preceding sound duration than in AAF, MGB responses also follow a saturating exponential function. We argue that the latter could perhaps become even more evident if probed with smaller sound durations (i.e., less than 50 ms), as neurons in earlier relays of the auditory pathway tend to have shorter time constants than cortical neurons. Thus, it could be that authors tested sounds too short compared to MGB time constants, resulting in a directly saturated effect. (**A.**) Adaptrans reproduces the decrease in offset response amplitude as a function of preceding fall-ramp duration. The experimental setup *(left)* simply consisted of a convolution of an

AdapTrans kernel over 1D fall-ramp stimuli; the maximum offset responses over the whole time course were then reported and the AdapTrans parameters were empirically optimized to best fit biological data *(right)* (**B**.) Similarly, offset response amplitude increases as an exponential function of preceding sound duration, measured in regular spiking (RS) and fast spiking (FS) neurons. The simulation setup was the same as for the previous panel, except for the stimuli, which were steps of infinite ramp and variable durations. (**C**.) Biological responses to multi-frequency stimulus can be accurately replicated *(right)* by a simple model *(left)* built upon the frequency-wise AdapTrans scheme. The latter processed each spectral component with its own version of AdapTrans, each with 3 parameters ($a_{ON}$, $a_{OFF}$, $w$); after a rectification (ReLU) stage, ON and OFF traces for each spectral component were then weighted and combined into a single final channel (3 bands × 2 polarities = 6 additional parameters).

the biological plausibility of our model, this simulation also illustrates that our framework easily permits to test any property. For instance, we can easily extract the latency of offset responses as a function of ramp duration (see S1 Fig). To our knowledge, it has not been tested in previous experiments. We hope future studies will explore whether the predictions of our model are correct in this case.

## Dependence of OFF responses on preceding sound duration

Another well-established property of auditory offset responses in the mammal brain is their dependency on preceding sound duration. This dependency takes the form of a saturating exponential [11, 12, 20]. In order to test whether AdapTrans can reproduce this relationship, we fed the OFF channel of our filter with auditory stimuli consisting in binary step functions. Fig 4B shows the maximum amplitudes of the responses to the offsets of these stimuli. These values can also be computed from AdapTrans OFF impulse response. If $T$ is the number of discrete timesteps of the step stimulus, the offset response amplitude at sound termination is:

$$A_{OFF}(T, a, w) = (1 - a)\sum_{d=1}^{T} a^{d-1} \tag{13}$$

Note that $A_{OFF}$ is an exponential function depending only on $T$ and $a$ (not $w$) and saturating to a maximum value as the preceding stimulus gets longer (i.e., in the limit of infinite $T$). Indeed, longer step stimuli can build up bigger exponential moving averages (left part of AdapTrans kernels in Fig 1A), leading to a bigger difference between present and past values and therefore to a bigger response when it stops. To further demonstrate the ability of AdapTrans to capture this biological property, we fitted this function on data collected in two auditory areas of the mouse cortex [12]. Results are reported in Fig 4B. The predictions of our model match remarkably well with experimental data ($R^2 = 0.94$). This is notably the case for data collected in the Medial Geniculate Body (MGB, in green) which are less favourable to offset responses than measurements made in cortical areas such as the Anterior Auditory Field (AAF, in orange). [12].

## The multi-frequency processing scheme of AdapTrans is in line with experimental data

Auditory neurons in areas MGB and AAF of the mouse auditory cortex can detect the offsets of different frequency components within complex structures of sound stimuli [12]. To test whether our approach can reproduce these results, we designed an artificial binary stimulus composed of three spectral bands with activations set to 1 and turned down to 0 at different time instants (respectively 0.3, 0.4, and 0.5 s in simulation time, $dt$ = 1 ms, see Fig 2) and filtered through AdapTrans, using different ON and OFF time constants (parameter $a$) and $w$ for each frequency band. Each output channel was then rectified and the final neural response

was readout at each timestep through weighted summation of each frequency-polarity bin, following the addition of a bias accounting for baseline neural activity. This simple 15-parameter model was finally fitted to reproduce experimental data averaged over a large number of thalamic and cortical neurons.

The resulting model activity closely matches the experimental data (see Fig 4C). This observation strongly supports the use of learnable time constants for each frequency band and polarity of the cochleagram, rather than the use of the raw sound waveform envelope. This experiment also demonstrates that AdapTrans can be used as a foundation or building block for larger models of audition.

## AdapTrans filtering enhances the neural fitting performances of a large spectrum of models

In order to test whether our framework can constitute a valuable extension to gold-standard models of auditory processing, we trained several computational pipelines, with and without AdapTrans, on electrophysiological data collected in auditory areas [41–43] (see Materials and methods, Datasets). This training was performed with supervision using back-propagation in order to predict new single-unit activity (see the 'Task and evaluation of performance' section). Fig 2 gives an overview of the proposed processing pipeline when the AdapTrans filters are used. The architecture of the different models of the auditory pathway which were implemented is detailed in the Methods (see the 'Computational neuron models' section). For each model and each dataset, we report in Table 3 the (cross-validated) raw and normalized Pearson correlation coefficients ($CC_{raw}$ and $CC_{norm}$) between the predicted and groundthuth neural responses, averaged over all units of the dataset.

**Table 3. Performances of the models with various prefiltering conditions on the datasets.** Correlation coefficients (CC) and normalized correlation coefficients are given in %. **Bold font** indicate the best prefiltering for a given model backbone, while underlined scores indicate the best among all models on a given dataset. All bold entries were deemed statistically significant with unilateral paired t-tests (best versus second best, $p = 0.05$).

| Model | | Dataset | | | | | | | |
|---|---|---|---|---|---|---|---|---|---|
| | | NS1 | | Wehr | | NAT4 | | | |
| | | | | | | PEG | | A1 | |
| Backbone | Prefiltering | $CC_{raw}$ | $CC_{norm}$ | $CC_{raw}$ | $CC_{norm}$ | $CC_{raw}$ | $CC_{norm}$ | $CC_{raw}$ | $CC_{norm}$ |
| L | None | 35.9 | 49.4 | 16.1 | 16.4 | 25.5 | 36.0 | 31.6 | 43.7 |
| | IC Adaptation | 42.7 | 58.8 | 30.7 | 31.2 | 18.3 | 24.6 | 23.2 | 31.1 |
| | AdapTrans | **43.1** | **59.2** | **31.0** | **31.6** | **30.4** | **41.6** | **37.0** | **50.1** |
| LN | None | 33.7 | 46.6 | 15.8 | 16.0 | 28.4 | 39.8 | 34.6 | 47.8 |
| | IC Adaptation | 42.3 | 58.1 | 26.4 | 26.9 | 17.6 | 23.7 | 23.6 | 31.8 |
| | AdapTrans | **43.3** | **59.6** | **27.3** | **27.7** | **31.2** | **43.3** | **38.0** | **52.3** |
| NRF | None | 43.7 | 60.4 | 16.5 | 16.6 | **30.3** | **41.6** | 36.9 | **49.8** |
| | IC Adaptation | **46.6** | 64.2 | 26.3 | 26.7 | 26.1 | 36.0 | 26.8 | 35.5 |
| | AdapTrans | **46.6** | **64.3** | **26.4** | **26.9** | 30.0 | 40.5 | **37.2** | 49.6 |
| DNet | None | 34.0 | 46.7 | 13.7 | 13.9 | 32.6 | 45.3 | 39.0 | 53.3 |
| | IC Adaptation | 44.1 | 60.9 | 24.5 | 25.0 | 27.4 | 37.9 | 32.4 | 44.0 |
| | AdapTrans | **44.4** | **61.3** | **25.5** | **26.0** | **36.3** | **50.8** | **43.1** | **59.3** |
| 2D-CNN | None | 47.2 | 65.1 | 17.2 | 17.5 | 35.2 | 48.5 | 41.7 | 56.8 |
| | IC Adaptation | <u>**48.6**</u> | <u>**67.1**</u> | 26.1 | 26.6 | 29.7 | 40.8 | 37.0 | 50.6 |
| | AdapTrans | 48.3 | 66.7 | **26.9** | **27.4** | <u>**37.9**</u> | <u>**52.9**</u> | <u>**44.3**</u> | **60.9** |

Overall, using AdapTrans significantly increases fitting performances. As an illustration, Fig 5 shows the predicted responses for different neurons sampled from the three datasets. Predictions using our approach are qualitatively and quantitatively better (see the associated correlation coefficients on the upper-right parts of the panels). On average, this increase in performances with AdapTrans reaches 0.059 in $CC_{raw}$ and 0.117 in $CC_{norm}$ (mean absolute delta between all baseline and AdapTrans models across datasets). Importantly, for each dataset, our method consistently provided better correlation coefficients (see the bold and underlined scores in the tables), with very few exceptions. This was true for both correlation coefficients ($CC_{raw}$ and $CC_{norm}$). Increases in CC were generally higher for smaller models (L, LN) but significant improvements were also found for NRF, DNet and 2D-CNN models (see the values in bold). Interestingly, we noticed that the improvements brought by AdapTrans on the "Wehr" dataset were close to 50%, even among the more sophisticated models (DNet, 2D-CNN). This dataset was the only one containing whole-cell current clamp recordings and was characterized by a very low inter-trial variability in neural recordings. Despite this high quality, it was associated with rather poor fits. It could be that this nature of recording is intrinsically more difficult to capture. Additionally, simpler models (L, LN) equipped with AdapTrans outperformed the more sophisticated ones, hinting at a possible overfitting problem. Anyhow, AdapTrans brings a nonlinearity that seems sufficient for the L/LN models to thrive on this particular set. In any way, the decomposition of the cochleagram into ON and OFF spectrograms performed by our approach greatly improve the predictions of neural responses in this case.

To determine whether the improvements observed in Table 3 are specific to some units or rather uniformly distributed across the neural populations, we provide in Fig 6 scatter plots with the $CC_{norm}$ values obtained with (y-axis) and without (x-axis) AdapTrans. For each dataset, this is done with the 2D-CNN model (which is the best performing baseline model across all the data) and also for the LN approach because of its simplicity, interpretability and its frequent use in studies modeling auditory processes. Scatter plots associated with the other backbones are provided in the supplementary materials (see S2 Fig). We can observe that most neural units, and especially the most reliable ones denoted by a $CC_{max}$ close to 1, are better fitted using AdapTrans. AdapTrans was notably beneficial for all neurons of the Wehr dataset (n = 20) and for a majority of units in the NS1 (70/73) and NAT4 dataset (516/777 for A1 and 229/339 for PEG).

In order to test whether AdapTrans provides a pre-filtering of the input spectrogram that is more beneficial to the neural response fitting task than previous approaches, we also trained downstream models using a re-implementation of the IC adaptation method described by [33] and running under our PyTorch framework. This re-implementation consisted of a cochleagram prefiltering step with the AdapTrans ON channel, frequency-dependent time constants initialized logarithmically but not learnable, a parameter $w = 1$ (transient information only), and a ReLU rectification. The associated performances are provided in Table 3. The IC Adaptation prefiltering greatly helped models on NS1, the original dataset on which it was developed. It also provided substantial improvement on the Wehr dataset, but in a minor extent than AdapTrans. Importantly, it failed on both NAT4 A1 and PEG datasets, doing worse than the baseline, whereas AdapTrans rather improved correlation scores. We explain this failure of IC Adaptation and success of AdapTrans by the prevalence of OFF responses in these datasets. Because IC Adaptation is in fact the half-wave rectified ON AdapTrans channel, sound offsets are therefore not present anymore in the stimulus spectrograms given to downstream models, and as a result the latter struggle to retranscribe OFF responses. This phenomenon is exemplified in Fig 5. These latter results highlight the need of the more general and flexible prefiltering framework provided by AdapTrans (i.e., IC adaptation does not segregate ON and OFF

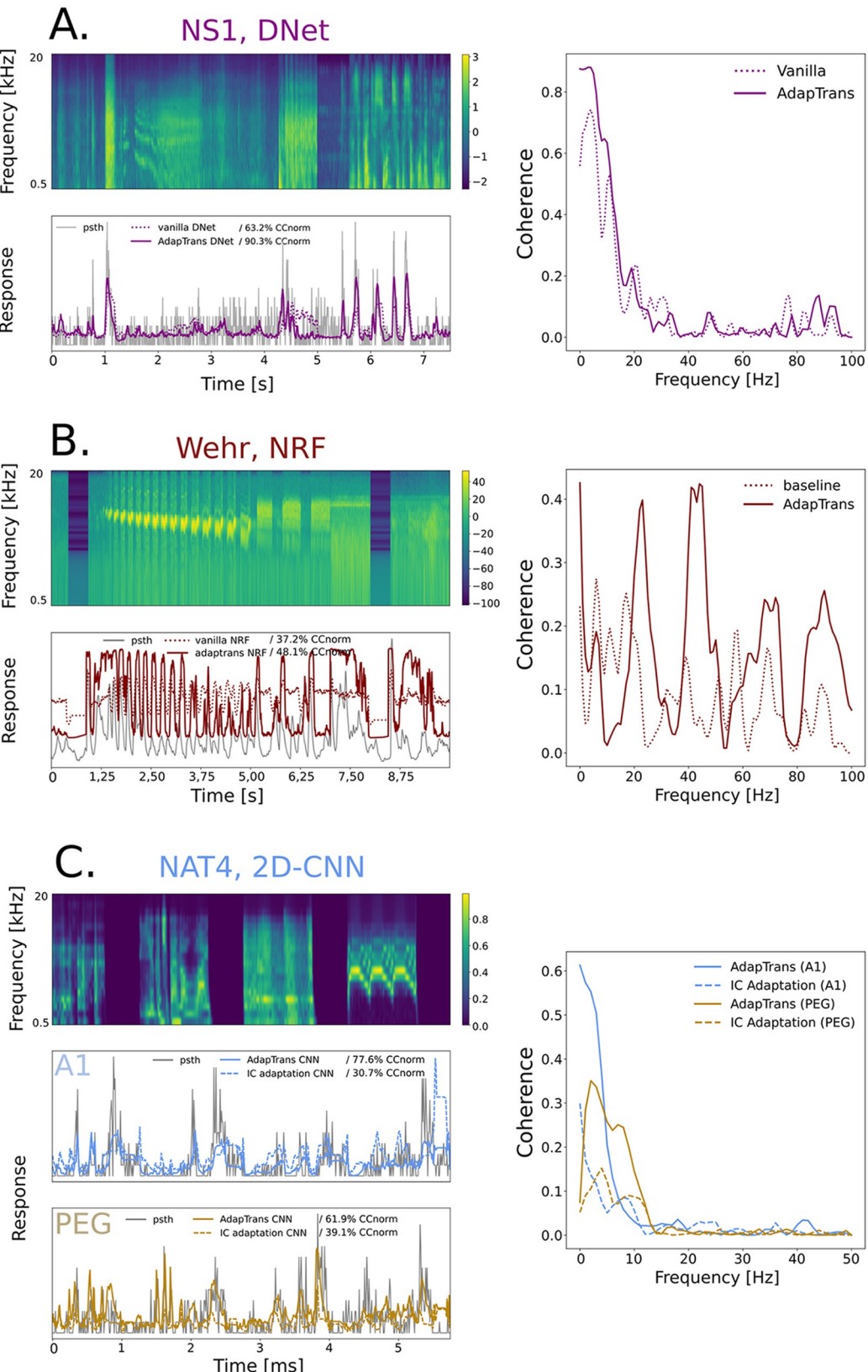

**Fig 5. Various deep learning models fitting auditory neural responses: Model prediction versus actual neural activity.** *Top left of each panel*: stimulus spectrogram from the test set, used to evaluate the performances of each model. *Bottom left*: comparison of groundtruth and model predictions, with and without AdapTrans, or with IC Adaptation. Indicated percentages represent the normalized correlation coefficients of the corresponding models for these clips. *Right*: coherence functions of the models estimated on the entire test set of the neuron. (**A.**) DNet model predictions on NS1 Dataset (ferret A1 spikes), unit #42. AdapTrans models often better capture transient variations of activity, in particular the height of peaks. The major contribution of AdapTrans in terms of coherence seems to lie in the low frequencies (0–10 Hz). (**B.**) Network Receptive Field (NRF) model benefits from AdapTrans on Wehr Dataset (rat A1 potentials), unit #5. In contrast to its baseline counterpart, the AdapTrans-enhanced model was able to better predict neural activity during inter-stimulus intervals and sustained stimulation. In a similar fashion as the DNet model on the NS1 unit above, AdapTrans primarily increases the coherence of slow spectral features (0–10 Hz). (**C.**) 2D-CNN model on NAT4 Datasets (ferret A1 and PEG spikes), units #359 and #29. This example is an illustration of the presence of OFF responses in these datasets, and shows that the latter are not properly captured by the IC Adaptation model due to only having the ON spectrogram channel. Similar to the example from NS1, the coherence improvement seems concentrated at lower response frequencies, suggesting that the latter are associated to transient responses, regardless of their polarity.

responses, discards all sustained information, and does not treat adaptation time constants as free, learnable parameters).

In brief, AdapTrans was shown to provide a simple first preprocessing layer that performs an efficient decomposition of input stimuli into dual features. The use of ON and OFF spectrograms almost systematically improves the predictions of neural responses, despite the wide variability in the datasets which contain different temporal (5–10 ms) and spectral (18–49 bins) resolutions of input cochleagrams, different animal species (rat, mouse, ferret) and areas (A1, AAF, PEG), different natures of recordings (spike sorted extracellular activity, patch-clamp).

## Coherence analysis

To complete our analyses, we also computed the coherence between the measured and estimated neural responses (see e.g., [44]). For each frequency, this metric takes values between zero (no correlation between measured and estimated response) and one (perfect correlation at this frequency). As an illustration, coherence values obtained with and without AdapTrans for the 4 neural units discussed above (units #42 of NS1, unit #5 of Wehr and units #359 and #29 of NAT4) are also shown on Fig 5. For all these units, coherence is generally higher for predictions based on AdapTrans, notably at low temporal frequencies. To quantify this effect, we show in Fig 6 the average coherence (across neural units) for each dataset and confirm that AdapTrans coherence is above the baseline for low frequencies (0–10 Hz), which could be associated to transient peaks of activity, such as ON or OFF responses.

## Parameters of the model after optimization follow a biologically plausible distribution

Here we show that AdapTrans parameters learnt through the neural response fitting process described above (see the 'AdapTrans filtering enhances the neural fitting performances of a large spectrum of models' subsection) converged towards values that are in line with electrophysiological recordings.

Fig 7 provide the average distributions (across neural units, models) of the $w$ and $a$ parameters. In general and for all datasets, it is interesting to observe that optimal values for $w$ remain high (i.e., in the [0.5, 1] interval) in all the frequency bands (see panel B). It suggests that neural responses in auditory areas are more strongly modulated by the transient rather than by the sustained properties of the input sounds. Time constants (i.e., parameter $a$, see panel A) remain overall close to their initialization values which were directly inspired from biological findings (see Eq 5), in the order of the hundred of milliseconds. It is nonetheless remarkable to observe that the optimization process led to greater values for the OFF channel rather than for

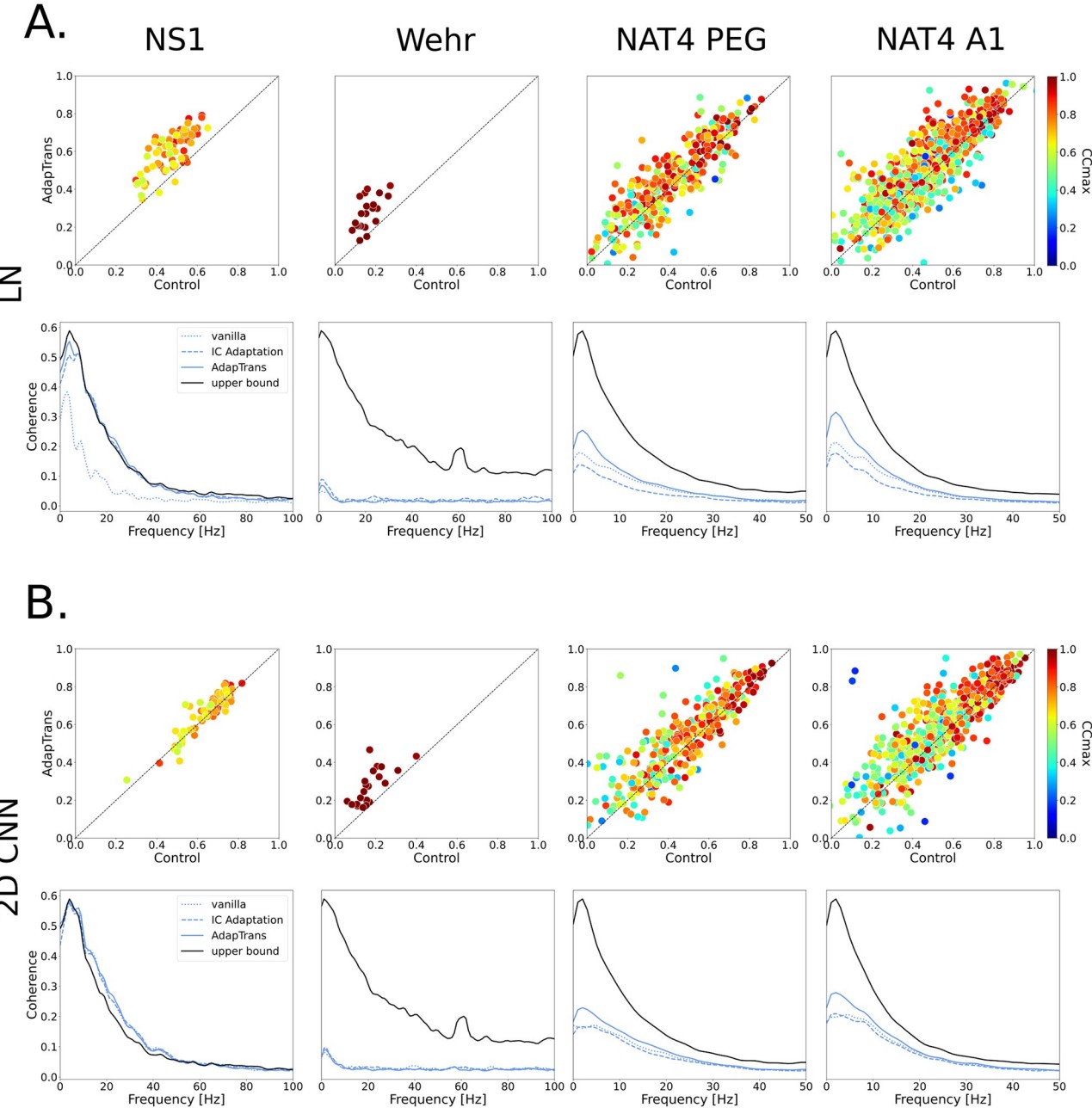

**Fig 6. Comparison of the neural fitting performances with and without the AdapTrans filters.** (**A**) (*top panels*) Comparative scatter plots showing the normalized correlation coefficients between model predictions and measured neural responses, for the LN backbone. The x-axis shows performances of the standard LN-model while the y-axis provides the performances when this model is combined with AdapTrans. Color denotes each unit's intrinsic variability, as the $CC_{max}$ coefficient ([33, 60]) A $CC_{max}$ of 0 means that the unit is unreliable to the point of only producing noise, while a value of 1 means that its responses are completely clean. We can see that AdapTrans seems to help reliable units in particular. (*bottom panels*) Test coherence function averaged across all units of the dataset. It can be seen that AdapTrans allows a better fit of lower response frequencies. (**B**) Same plots for the 2D CNN model. If the improvement in terms of $CC$ brought by AdapTrans is perhaps less clear than for the LN model as performances saturate (possible through ceiling effect), the scatter distribution remains skewed towards the upper left corner. For NS1, the coherence functions measured can even surpass the upper bound, which we attribute to the fact that the estimation of the coherence function a statistic and inexact process. **N.B**. Coherence values can be obtained through an estimate of power spectral density (PSD), which implies non-exact, statistical methods. In addition, the coherence upper bound was calculated with a finite number of combinations to create PSTHs. As a result, it can be that a highly performing model can surpass the upper bound, by chance.

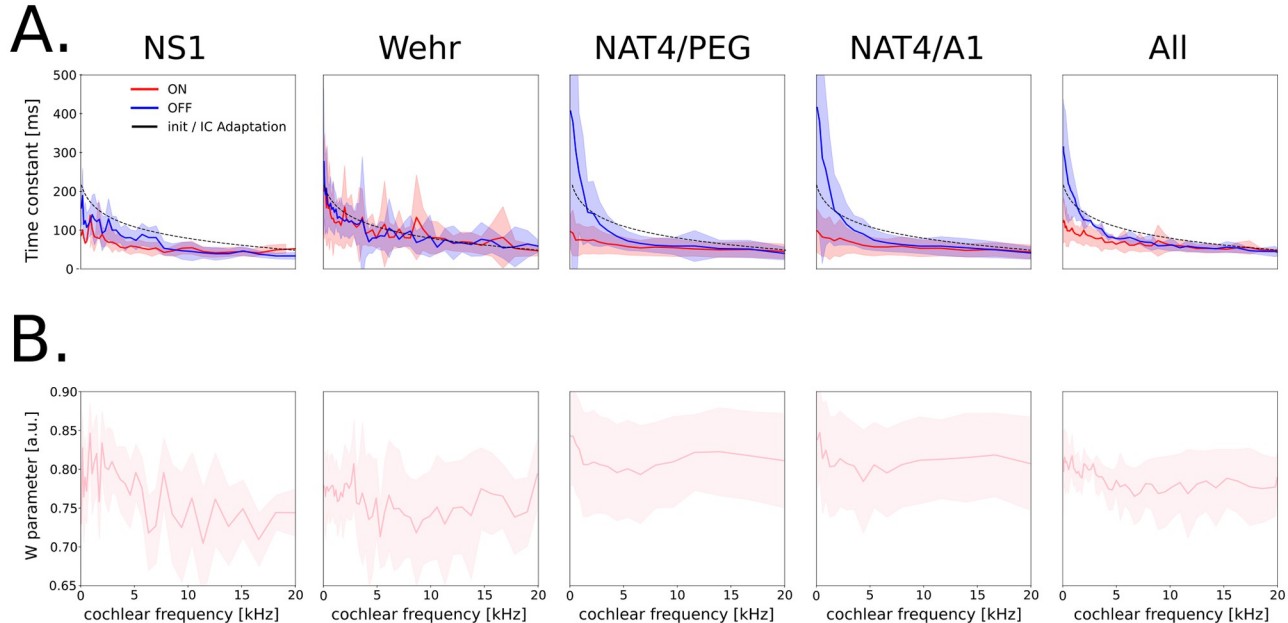

**Fig 7. Distribution of AdapTrans parameters learned during the neural fitting process as a function of frequency.** Values were averaged across neural units and models. The errorbars correspond to ± the standard deviations across neural units, except for the "**All**" subpanel where it is across datasets. (**A**.) ON and OFF time constants. These constants decrease with frequency. Values obtained for OFF responses are almost systematically higher than those observed for ON responses. (**B**.) $w$ remains at high values even after learning, thus favoring transient over sustained responses.

the ON channel. This was true in the three datasets and for all the models. This result is in line with the findings of previous electrophysiological studies ([13, 63]) and could reflect statistical differences between the onsets and offsets of natural sounds ([18, 64]).

Comparing these distributions between datasets can be insightful about how adaptation properties vary depending on the animal species, brain area, or behavioural state. In particular, we noticed that time constants from PEG and A1 neurons of the NAT4 dataset were higher than those estimated from NS1, despite the fact that the associated recordings were performed in the same animal model (ferret). This faster adaptation for NS1 could be explained by the behavioural state (ferrets were awaken in NAT4 and anesthetized in NS1) as suggested by [65]. At the opposite, the $w$ parameter controlling for the trade-off between steady and transient signals followed similar distributions in NS1 and NAT4, with a preference for transients. As a result, we can hypothesize that behavioural state alters the time scale of filtering processes performed by auditory neurons, but not their nature (i.e., high-pass or low-pass). Finally, we found similar time-constant distribution in rats (Wehr) and ferrets (NS1, NAT4) A1 neurons, which might reflect the convergent evolution of the early auditory cortex in these two mammal species. However, OFF time constants were not higher than their ON counterpart in the case of the Wehr dataset.

### Further performance improvements with an augmented AdapTrans scheme and neural population training

In this final subsection, we investigate how two simple improvements in our approach can further push its performances at neural response fitting.

**Adding the raw spectrogram to AdapTrans.** AdapTrans is based on bipolar spectrograms with per-frequency and per-polarity adaptation mechanisms. Because it is possible that

**Table 4. Performances of the 2D-CNN model with an AdapTrans scheme augmented by the raw, unadapted spectrogram as a third input channel.** Correlation coefficients (CC) and normalized correlation coefficients are given in %. **Bold font** indicate the best prefiltering for a given model backbone.

| Model | | Dataset | | | | | | | |
|---|---|---|---|---|---|---|---|---|---|
| | | NS1 | | Wehr | | NAT4 | | | |
| | | | | | | PEG | | A1 | |
| Backbone | Prefiltering | $CC_{raw}$ | $CC_{norm}$ | $CC_{raw}$ | $CC_{norm}$ | $CC_{raw}$ | $CC_{norm}$ | $CC_{raw}$ | $CC_{norm}$ |
| 2D-CNN | AdapTrans | 48.3 | 66.7 | 26.9 | 27.4 | 37.9 | 52.9 | 44.3 | 60.9 |
| | Adap+raw | **48.6** | **67.1** | **28.0** | **28.5** | **38.8** | **54.2** | **45.6** | **62.8** |

some neurons along the auditory pathway do not adapt to incoming stimuli, we explore here a new version of our approach that explicitly takes into account this hypothesis. Theoretically, AdapTrans parameters can be learned such that they implement the identity transform –and therefore no adaptation– when necessary but there is no strong guarantee that it happens in practice. In this augmented version, the raw (and thus unadapted) spectrogram is concatenated with the ON-OFF spectrograms. This 3-channel (adap-ON, adap-OFF, raw) spectro-temporal representation is then used as an new input for the downstream models. We tested this on the most consistent model across datasets, that is the 2D-CNN; the $CC_{raw}$ and $CC_{norm}$ obtained are shown on Table 4. We can observe that this simple modification systematically improve performances.

**Predicting neural population activity.** A recent study suggested that it can be beneficial for computational models with high parameter-counts to predict the activity of several neurons simultaneously [30]. Such an approach strongly reduces the number of effective degrees of freedom used for each unit, speeds up the training and boosts performances by learning a joint and thus more meaningful embedding (i.e. representation) that is less prone to overfitting. As can be seen above, AdapTrans parameters do not seem to vary greatly among units of the same dataset, so we investigated whether using such an approach as an early processing step of a population model could be beneficial.

This was done by equipping a model of population activity with AdapTrans, and training it using the same pipeline as our previous single unit models. The only difference between the single unit and population models was the number of output units (respectively 1 and $N$, $N$ being the total number of valid units in the target dataset). The loss was still given by the mean squared error (MSE) between the predicted and measured signals, now with an averaging operation across output units. Note that contrarily to [30], we did not train the whole processing pipeline in two steps –backbone and readout– but all at once. Because this approach requires the responses of neural units to the same stimuli, which are not available in the case of Wehr dataset, we only report the performances on NS1 and NAT4 in Table 5 below. We applied this approach on the 2D-CNN model because of its high parameter count and overall better performances, but also because it was the model on which this technique was originally proposed [30]. We find consistent and significant improvements across all datasets, further pushing the limits of auditory neural response fitting, and showing that AdapTrans is highly compatible with this efficient optimization process.

As a conclusion, we showed here that AdapTrans can be further improved with simple additions such as the incorporation of a unadapted version of the spectrogram, and can also enhance the capabilities of computational models of neural populations.

## Discussion

In this paper, we describe a new general, descriptive model of neural responses in the mammal auditory pathway. Our model is composed of two linear filters that capture the sustained and

transient properties of auditory inputs (see [38] for an illustration of this concept in the visual domain). Contrarily to most previous modeling works, it takes into account both the ON and OFF responses and processes them independently within each frequency band (see the 'Model' section). This segregation is justified by the results of previous studies which established that separating the ON and OFF systems improves sensory coding and actually provides a better code for extracting meaningful information by a downstream decoder (see e.g., [1]. We demonstrated here that our framework accurately reproduces known properties of neural responses in the auditory cortex such as the dependence of OFF responses on the stimulus fall time and on the preceding sound duration (see the Dependence of OFF responses on stimulus fall time and 'Dependence of OFF responses on preceding sound duration' subsections). By combining data from numerous studies collected in different animal models and auditory areas (see Table 2), we also demonstrated that AdapTrans almost systematically improves neural fitting performances of a large gamut of models of the auditory pathway (higher correlation scores were observed in 18 cases over 20), going from simple linear models to state-of-the-art convolutional neural networks (see Table 3 and Figs 2 and 3). The overall increase of normalized correlation scores was above 0.117 across all datasets tested (see the 'Results' section). Using a coherence analysis, we also showed that our approach improves neural fitting within a large frequency band and notably at low response frequencies, possibly associated with transient peaks of activity and action potential generation. Finally, we showed here that AdapTrans can be further improved with simple additions such as the incorporation of an unadapted version of the spectrogram, and can also enhance the capabilities of computational models of neural populations (as in [30]). This approach significantly speedups training (training time becomes almost independent of the number of neurons!) and boosts performances (see Table 5) and should therefore be used in future work. Importantly, except for early stopping and batch normalization, we did not use any other form of regularization, nor parameterization. In preliminary tests, we explored whether weight decay (a L2 penalty readily implementable in our PyTorch setup) upon spectro-temporal weights could improve performances but this manipulation had little impact on the results (see S2 Table). This observation is in line with the results reported in [30], obtained without such optimization constraints.

One of the interests of our framework is that the optimal distribution of the model parameters ($\tau$ and $w$) can be directly derived from experimental data through the neural fitting process (see the 'AdapTrans parameters follow biologically-plausible distributions after learning' section). For all the tested models, we encouraged time-constants (parameter $\tau$) to decrease as frequency increase (see Fig 7) through an initialization in line with previous biological findings [36]. Interestingly, in addition to remaining close to that initial distribution, we found that optimal time constants were significantly higher for OFF than for ON responses in 3/4 of the tested datasets. This result is in agreement with previous measurements in the ferret [63] and cat [13] auditory cortices and might reflect an efficient encoding of statistics in natural sounds

**Table 5. Model performances (CC) obtained through fitting whole population activity, rather than single unit activity. Bold fonts** indicate the training mode (single / population) for each of the control and AdapTrans versions.

| Model | | | Dataset | | | | | |
|---|---|---|---|---|---|---|---|---|
| | | | NS1 | | NAT4 | | | |
| | | | | | PEG | | A1 | |
| Backbone | Prefiltering | Mode | $CC_{raw}$ | $CC_{norm}$ | $CC_{raw}$ | $CC_{norm}$ | $CC_{raw}$ | $CC_{norm}$ |
| 2D-CNN | None | Single | 47.2 | 65.1 | 35.2 | 48.5 | 41.7 | 56.8 |
| | | Population | **47.9** | **66.2** | **36.0** | **50.5** | **43.4** | **60.1** |
| | AdapTrans | Single | 48.3 | 66.7 | 37.9 | 52.9 | 44.3 | 60.9 |
| | | Population | **50.0** | **69.2** | **39.2** | **55.2** | **46.4** | **64.5** |

where offsets are usually slower and less salient than onsets [18, 64]. In addition, our approach also predicts that the optimal *w* ranges between 0.5 and 1, which confirms that auditory responses are better captured when both the transient and sustained parts of auditory inputs are taken into consideration, in line with biological findings (see e.g., [9]). Finally, we also found that this optimal *w* remains stable across frequencies. Thus, we predict that the relative contribution of transient versus sustained responses is frequency independent. To our knowledge, this hypothesis has never been tested in animal models. We hope that future experimental works will explore this lead.

Previous studies proposed models that shared properties with our framework [27, 33, 34]. [33] modeled adaptation to mean sound level in the auditory midbrain using a high-pass filter with frequency-dependent time constants. However, they did not segregate between the ON and OFF pathways and their model can thus only capture whether the sound intensity has changed in a given frequency band but not whether this modification reflects an increase or a decrease. In addition, their model only considers transients and completely discards the sustained properties of the auditory inputs. As we saw it, our framework is more general as it segregates ON and OFF responses and considers both sustained and transient properties whose relative contributions are controlled with the *w* parameter (see Eq 1 in the '"AdapTrans" model of auditory ON-OFF responses and adaptation' section). Note that fixing this parameter to one and discarding the OFF channel actually brings our model back to the IC adaptation model proposed by [33]. It is thus not surprising that the correlation scores obtained with AdapTrans are (almost) systematically better than those observed with IC adaptation (the only exception being for the 2D-CNN model on NS1, see Table 3).

[27] proposed a model based on divisive normalization that takes into account ON and OFF responses, although these two channels are ultimately merged together. Their method uses the 1D sound envelope and thus does not process each frequency band separately. Adap-Trans is based on subtractive normalization and is more general as it is applied on the 2D sound spectrogram of the auditory inputs and relies on parameters that are frequency dependent. By learning from experimental data, our approach permits to optimize the relative contributions of the ON and OFF channels for each frequency band, whereas this relative contribution is fixed and global in [27]. Also, our method is more biologically plausible as it keeps separated the ON and OFF channels along the entire hierarchy of the auditory cortex.

The model proposed by [34] also incorporated adaptation mechanisms in different frequency bands which better accounts for transient and sustained stimulus features than global adaptation. According to [66], such nonlinear forms of adaptation are paramount for better encoding models. However, neither of these approaches included polarity tuning in the form of ON and OFF responses.

There are other biological properties that our approach cannot properly capture in its current form. For example, some asymmetries (i.e., in amplitude, latency, spectral tuning) commonly observed between ON and OFF STRFs do not seem to emerge implicitly from training on a machine learning task such as response fitting. This issue could however be easily resolved by explicitly parameterizing ON and OFF weights subsequent to AdapTrans altogether. Another example is given by the non-instantaneous buildup times of ON/OFF responses which are currently not well predicted by our model. This could also be easily sorted out by replacing the initial Kronecker delta function (see Fig 1A) by a growing exponential, at the cost of one extra parameter (i.e., the time constant of this added exponential part). These leads will be explored in future works from our team.

Altogether, we present here a unifying framework that encompasses previous approaches ([27, 33]) and permits to improve our understanding of computations performed in the

mammal auditory pathway. AdapTrans can serve as a transparent primitive and a key layer of computation [29] to account for the broad range of neural response patterns observed in the mammal auditory cortex such as ON, OFF, ON-sustained-OFF, inhibition-OFF, among others [9, 17, 27, 32]. Such a wide algorithmic-level model (in opposition to Marr's implementation level, see [67]) has been lacking so far in the field. We hope it will inspire new modelling works at the mechanistic level in order to clarify what might be the biophysical implementations of the computations described here (in the terminology of [68] and [69]). Beyond audition, we think our approach could also account for processing in other sensory modalities and notably in the visual system where the segregation between ON and OFF luminance information has been documented at multiple stages of processing, starting from the retina [2, 3]. Applying AdapTrans in a pixel-wise manner to temporal image sequences could thus improve the bio-plausibility of existing models of the early visual pathway. In terms of applicative researches in artificial intelligence, the integration of our framework within state-of-the-art deep learning models could improve performances in numerous sensory-based tasks, such as sound classification or object detection. In this sense and in order to facilitate the reproduction of our results and the use of our approach in future studies, we provide all the data, models, processing and pre-processing codes used in the present work at the following github repository. We also provide user-friendly Pytorch Dataset classes for each source of data.

## Supporting information

**S1 Note. Mathematical derivation of AdapTrans properties.** We show here how to mathematically derive analytical formulas for the transfer functions, magnitude-frequency curves, poles and zeroes, as well as a recursive formulation of our AdapTrans filters.
(PDF)

**S1 Code. Pseudocode demonstration.** We demonstrate here the simplicity of our code, hosted at the url https://github.com/urancon/deepSTRF, by showing its basic Pytorch flow. As the code is continuously updated and has been simplified for pedagogical purposes, minor differences can exist between the following snippet and the online version.
(PY)

**S1 Fig. AdapTrans offset response latencies as a function of ramp duration.** The simulated experimental setup described in Fig 4A allows to extract the AdapTrans offset response latency as a function of the ramp duration. S1 Fig below shows this relationship for different pairs of $w$ parameter and OFF time constant (tau). Our model predicts that latency increases as fall ramps become less abrupt. To our knowledge, this has never been tested by electrophysiologists. We hope that future experimental studies will address this question.
(PDF)

**S2 Fig. Scatter and coherence plots for the other models.** We present below the scatter and coherence plots for the L, NRF and DNet models on all datasets.
(PDF)

**S1 Table. Risks of using a fixed test set for cross-validation.** In [50], a fixed data subset was held-out during model development, and "opened" back only for testing models, while various splits of the remaining data were used for training and validation. This strategy might be risky if the selected dataset is significantly easier or harder than the training and validation sets, and results in over or under-estimated performances. The table presents Test performances of models depending on the cross-validation methodology. Models and the rest of the training pipeline were identical for both conditions. Performance metrics are the correlation coefficient

and normalized correlation coefficient, in %.
(PNG)

**S2 Table. Influence of weight decay on model performance.** The auditory response fitting literature has a long tradition of explicit model regularization in the form of L1 or L2 penalty on STRF weights. However, initial models were fitted via zero-order algorithms such as boosting, and not gradient descent (as in this study), an ubiquitous optimization algorithm which may display some intriguing implicit regularization [51]. Following innovations from the machine learning community, we have incorporated batch normalization in our models, and followed a rigorous fitting procedure with separate training and validation sets and an early stopping criterion. During our initial model development phase, we investigated whether the weight decay readily available in PyTorch could lead to better performances. We report here results of control models on the NS1 dataset. The table presents test performances as a function of the weight decay parameter used with AdamW optimizer in our setup. Models and the rest of the training pipeline was identical for both conditions. Performance metrics are the correlation coefficient and normalized correlation coefficient, in %.
(PNG)

## Acknowledgments

Our thanks go to Céline Cappe for her valuable insights on the mammalian auditory system, and to Corentin Nivelet-Etcheberry for his help on pre-processing the datasets and fitting the models.

## Author Contributions

**Conceptualization:** Ulysse Rançon, Timothée Masquelier, Benoit R. Cottereau.

**Data curation:** Ulysse Rançon.

**Formal analysis:** Ulysse Rançon.

**Funding acquisition:** Benoit R. Cottereau.

**Investigation:** Ulysse Rançon, Timothée Masquelier, Benoit R. Cottereau.

**Methodology:** Ulysse Rançon, Timothée Masquelier, Benoit R. Cottereau.

**Software:** Ulysse Rançon.

**Supervision:** Timothée Masquelier, Benoit R. Cottereau.

**Validation:** Timothée Masquelier, Benoit R. Cottereau.

**Visualization:** Ulysse Rançon.

**Writing – original draft:** Ulysse Rançon.

**Writing – review & editing:** Timothée Masquelier, Benoit R. Cottereau.

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
