## [Decision Letter · Decision Letter 0]

22 Mar 2024

Dear Mr. Rançon,

Thank you very much for submitting your manuscript "A general theoretical framework unifying the adaptive, transient and sustained properties of ON and OFF auditory neural responses" for consideration at PLOS Computational Biology.

As with all papers reviewed by the journal, your manuscript was reviewed by members of the editorial board and by several independent reviewers. In light of the reviews (below this email), we would like to invite the resubmission of a significantly-revised version that takes into account the reviewers' comments.

Dear authors,

As you will read, two experts in the field have reviewed your manuscript and expressed enthusiasm for your contribution. This said they both raised major issues that need to be resolved prior to publicaiton. From reading their comments, it appears that you might have also misrepresented the work in prior publication and this will need to be fixed. I also very much like the suggestion of reviewer 1 who is encouraging you to use a coherence calculation for estimating the goodness of fit of your model. Looking forward to your revised manuscript.

Best,

Frederic Theunissen

We cannot make any decision about publication until we have seen the revised manuscript and your response to the reviewers' comments. Your revised manuscript is also likely to be sent to reviewers for further evaluation.

Sincerely,

Frédéric E. Theunissen

Academic Editor

PLOS Computational Biology

Lyle Graham

Section Editor

PLOS Computational Biology

Dear authors,

As you will read, two experts in the field have reviewed your manuscript and expressed enthusiasm for your contribution. This said they both raised major issues that need to be resolved prior to publicaiton. From reading their comments, it appears that you might have also misrepresented the work in prior publication and this will need to be fixed. I also very much like the suggestion of reviewer 1 who is encouraging you to use a coherence calculation for estimating the goodness of fit of your model. Looking forward to your revised manuscript.

Best,

Frederic Theunissen

Reviewer's Responses to Questions

**Comments to the Authors:**

Reviewer #1: This study develops a new front end for central auditory encoding models that explicitly and separately models adapting on and off responses to changes in sound level. The authors incorporated this AdapTrans filter into several published models and tested it on several datasets from A1 and IC. In nearly every case, addition of AdapTrans improved model performance.

The study was well-motivated by previous work, and the authors were clearly very thoughtful in their approach. The fact that they tested multiple models and datasets is especially novel, and they rightly identify issues around the inability to compare previous studies because of methodological differences. While this work is quite interesting and worth publishing, there are places where the findings are presented in an overly-compressed way, which makes it difficult to fully assess the results. These issues should be addressable.

MAJOR CONCERNS

p. 12. Table 4 provides convincing quantitative evidence for the benefit of the AdapTrans filter, but it is not clear how performance differences are distributed across the set of neurons tested. A scatter plot comparing performance of two models for each neuron clarifies how the improvements were distributed across the populations. Comparing every combination of models might be too much, but a more detailed characterization, for examples, of the higher-performing models would help.

p. 14. Somewhat related, the characterization of the key new parameters in Fig. 6 is collapsed across models and datasets. Since data are collected from different species and difference behavioral states (awake vs. anesthetized) it is especially important to not collapse across these datasets. Either result would be interesting, whether a and w are consistent or different between datasets.

p. 13. While it is clear that the new filter improves performance, it is difficult to tell from the example data in Fig. 5 where/how they improve predictions. It does look like peaks in the predicted PSTHs are narrower for the AdapTrans model, which would be consistent with a model that captures transient dynamics more accurately. Can this be quantified? For example, the authors might consider measuring the coherence between predicted and actual PSTHs to see if the benefits are specific to temporal frequency bands. This would help show whether AdapTrans is capturing specific dynamics or just enriching the input space more generally.

LESSER CONCERNS

Minor: It would be helpful if the manuscript could include line numbers.

p. 3. “see the ‘Model’ section”. There does not appear to be a section explicitly called “Model”. “Computational Neuron Models”?

p. 4. (also p. 15 Discussion) The w parameter provides a creative way to include both sustained and transient information in the stimulus representation. An alternative would be to simply include a third filter bank that does not adapt. Is there a specific theoretical benefit to fusing them in a single filterbank? Numerical efficiency?

p. 5. Table 1. It is not immediately clear what the “F” “T” and “H” headings refer to.

p. 6. CNN model. An argument of the Pennington and David study was that performance of high parameter count models benefits substantially from fitting to activity of the entire dataset in a single model. Could this architecture have been implemented in the current study?

p. 8. “Optimization process.” Can the authors clarify if the AdapTrans-specific parameters (w, a) were fit simultaneously with the other model parameters? And separately for each neuron? The language, especially around Eq. 5 is a bit confusing. Does Eq 5 indicate the initial values of a (tau?) for all models fits?

p. 8. “Cross validation methodology” section. It appears that different test sets were used, but the number of trials varied across stimuli, and in some cases there was only one repeat. How was CCnorm computed for the single trial data?

p. 8. Table 2. Please double check numbers. Eg, for the NAT4 dataset, the number of units does not match the numbers on p. 7. Also “3-63 x 1-25” for the Wehr dataset is difficult to parse.

p. 10 “… offsets of different frequency components with a rich spectral stimulus” It is not clear what this means.

p. 14. “… previous studies” The authors might also consider work by Lopez Espejo et al (PLoS CB 2019) that incorporated short-term synaptic plasticity into LN models, which also could account for sustained + transient responses (though did not distinguish between on and off). Additionally, work by Gill et al (JCN 2006) incorporated cochlear adaptation into LN models.

Reviewer #2: This paper provides a model of neural responses in auditory cortex which builds on previous models, explicitly incorporating sustained and transient ON and OFF responses. The model provides better performance on some data sets than previous models. This is an interesting problem, and the authors have presented a nice new model which appears to be a useful step forward in pragmatic, biologically plausible computational modelling of neurons in mammalian auditory cortex. However, I have a number of concerns which make it hard to compare the performance of the new model with other similar models. I am also concerned that some claims are misleading or unjustified. Additional analyses are needed to address these concerns.

1. The authors introduce a new measure of prediction quality -- cc_norm that is normalised on a per-neuron and per-stimulus basis. This makes it hard to compare with previous work. The authors do not clearly explain why they did this. Using this measure makes it hard to understand whether the authors' models have been fitted correctly, and how these results compare with previous efforts. For each stimulus set they use, the authors should provide a direct comparison with the original authors' models, using the prediction quality metric used by the original authors. If the authors want to use a new metric, they should explain clearly what they believe are the advantages of the new metric, and provide quantitative justification for this.

2. The authors make comparisons between their model and a similar model which the authors refer to as Willmore et al (2016). However, this comparison is quite misleading because the "Willmore et al (2016)" model used here has several important differences from the model that was introduced in Willmore et al (2016). These differences are highly likely to affect the performance of this model, and so it is incorrect (and misleading) to present the authors' results as a test of Willmore et al's model. The authors say that they failed to replicate Willmore et al's results. This is likely explained by the differences between the model that they tested and the actual model used by Willmore et al. To fix this, the authors should implement Willmore et al's model accurately. The differences are listed below. Alternatively, they should make it clear that the model they are testing has some similarities with Willmore et al but is not the same model, and remove claims about failing to reproduce Willmore et al's results.

3. Similarly, the NRF and DNet models as implemented here are not identical to the models used by the respective authors. In particular, the use of Sohl-Dickstein's sum of functions optimiser was important for these models (https://github.com/Sohl-Dickstein/Sum-of-Functions-Optimizer). Again, this means that the authors should not claim that their results are representative of those models.

4. AdapTrans has key similarities with the model presented here as "Willmore et al (2016)" (hereafter MWEA for "modified Willmore et al"). Yet the two models perform very differently here, with AdapTrans doing very well, and the MWEA doing very poorly (being beaten by an LN model). Additional analysis is need to uncover what is going on here. Why does the MWEA fail to improve on the LN model? What are the key properties of AdapTrans that make it so much better than the MWEA? The authors need to unpick exactly what differences between the two models (and the original Willmore et al model) are responsible for the differences in predictive power.

5. The authors make the very surprising claim that regularisation is not required when fitting linear models to physiological data. This is at odds with previous literature, which the authors cite. I suspect that this is a mistake, and the authors methods are somehow introducing regularisation (e.g. by early stopping). This claim needs to be explained in much more detail, evaluated carefully to see if it is really correct, and, if so, it needs to be quantitatively justified.

Major concerns

Pg 6. The authors make surprising claims that regularisation is not needed for fitting linear models. The previous literature, which they cite, has usually found that regularisation is critical when fitting these models to physiological data sets. This deviation from previous work is controversial. The authors say that regularisation "did not necessarily prove beneficial in terms of performance in our setup". This is insufficient justification for a controversial claim. I think the authors need to re-investigate this, and establish whether it is really true that no regularisation is being used here. One possibility is that the authors are effectively using early stopping (which can be seen as a form of regularisation) by using too few epochs to train their models. The authors speculate that their use of gradient descent might make regularisation unnecessary. There is some literature on this idea (e.g. https://arxiv.org/abs/2009.11162). I think detailed quantitative analysis is needed here, to explain how the models are avoiding overfitting.

Pg 8. The authors normalise cc_norm on a per-stimulus basis as well as per-neuron. The authors justify this as follows: "we decided to normalize per neuron and per stimulus, as there can be a variability in the responses depending on both of these factors". It is true that cc_norm is going to depend on both neuron and stimulus; however, it is not clear why or if the stimulus-based variability should be excluded. In the limit of short stimuli, normalising on a per-stimulus basis will lead to excluding all of the stimulus-based variability, and this approach will therefore tend to exaggerate cc_norm values. I think the authors need to explain why they think this approach is superior. They also need to provide analysis to quantitatively justify this non-standard choice, by showing why this is better than the standard approach of normalising only on a per-neuron basis (or per-neuron and per stimulus class). An alternative would be to use previously published measures of prediction performance.

Pg 7. The use of an unusual measure of prediction performance makes it difficult to compare model performance with previous efforts. In order for the reader to be able to understand that the authors' models have been fitted correctly, and how they compare with previous efforts, for each previous data set used, the authors should provide direct comparisons with previous models used by the authors of that dataset, using the same measures of prediction performance as the original authors. It is fine to use the new measure as well, if it is justified as described above.

Pg 8. Calculating cc_norm via cc_half is no longer the best approach. It makes sense to use this measure for comparison with Willmore et al (2016), which used the same method. However, I would recommend using the methods in Schoppe et al (2016) for calculating cc_norm in general.

Pg 9. The authors claim that Rahman et al (2019) overestimated the cc_norm of models. Rahman et al used an approach where a dataset was held out during the whole process of model development, and only 'opened' once the final models had been selected and trained. This approach has a particular benefit: the model hyperparameters are not overfitted to the final test set. This means that this approach is likely than o

---

## [Decision Letter · Decision Letter 1]

29 Jun 2024

Dear Mr. Rançon,

We are pleased to inform you that your manuscript 'A general model unifying the adaptive, transient and sustained properties of ON and OFF auditory neural responses' has been provisionally accepted for publication in PLOS Computational Biology.

We also recommend that you consider the final remarks of Reviewer 2.

Best regards,

Frédéric E. Theunissen

Academic Editor

PLOS Computational Biology

Lyle Graham

Section Editor

PLOS Computational Biology

As you will read, your two reviewers agreed that you did a good job at replying to their comments. Please fix the fix errors for you final version.

Best,

Frederic Theunissen

Reviewer's Responses to Questions

**Comments to the Authors:**

Reviewer #1: The authors have done a good job addressing the concerns raised about the previous submission. The manuscript now provides a compelling argument for the generality of their new adaptation model across datasets and model backends.

Reviewer #2: The authors have done a very comprehensive job of responding to the comments of both reviewers. I think the manuscript is greatly improved as a result, and should be published in something close to the current form. I think the new finding that AdapTrans is particularly effective on the NAT4 (awake) data is particularly interesting, and I hope to see more investigation of this in future work.

A few errors remain which need to be fixed:

Table 3: Some of the underlining appears to be wrong – the best performing model on NS1 is 2D-CNN with IC Adaptation, but 2D-CNN with AdapTrans is underlined; on the Wehr dataset, the best performing model is Linear with AdapTrans, but NRF with AdapTrans is underlined.

Lines 949-952: This is not correct, based on the values in Table 3, and I think this sentence should be corrected. The best CCnorm on NS1 is given by 2D-CNN with IC adaptation, as correctly stated on lines 1293-1294.

Lines 1090-1097: Higher time constants (as shown in Figure 7A) imply slower adaptation. But the text here suggests that that the adaptation in NAT4 is faster than in NS1.

Minor comments

Line 119: “improve” should read “improves”

Line 121: “use” should read “used”

Line 399: “value” should read “value”

Line 501: “accross” should read “across”

Equation 11: Left hand side should read CCnorm not CCraw

Line 932: “wihch” should read “which”

Lines 1030-1031: “improve” should read “improves”

Line 1139: Insert period after “2023b)”

Line 1141, line 1223: “speedups” should read “speeds up”

**Have the authors made all data and (if applicable) computational code underlying the findings in their manuscript fully available?**

Reviewer #1: Yes

Reviewer #2: Yes

PLOS authors have the option to publish the peer review history of their article (what does this mean?). If published, this will include your full peer review and any attached files.

Reviewer #1: No

Reviewer #2: No

---

## [Editor Report · Acceptance letter]

25 Jul 2024

PCOMPBIOL-D-24-00268R1 

A general model unifying the adaptive, transient and sustained properties of ON and OFF auditory neural responses

Dear Dr Rançon,

I am pleased to inform you that your manuscript has been formally accepted for publication in PLOS Computational Biology. Your manuscript is now with our production department and you will be notified of the publication date in due course.

With kind regards,

Lilla Horvath
